# Control of immune ligands by members of a cytomegalovirus gene expansion suppresses natural killer cell activation

Ceri A Fielding[1†], Michael P Weekes[2,3†], Luis V Nobre[2], Eva Ruckova[4], Gavin S Wilkie[5], Joao A Paulo[3], Chiwen Chang[6], Nicolás M Suárez[5], James A Davies[1], Robin Antrobus[2], Richard J Stanton[1], Rebecca J Aicheler[1§], Hester Nichols[1], Borek Vojtesek[4], John Trowsdale[6], Andrew J Davison[5], Steven P Gygi[3], Peter Tomasec[1‡], Paul J Lehner[2‡], Gavin W G Wilkinson[1*‡]

[1]Division of Infection and Immunity, School of Medicine, Cardiff University, Cardiff, United Kingdom; [2]Cambridge Institute for Medical Research, University of Cambridge, Cambridge, United Kingdom; [3]Department of Cell Biology, Harvard Medical School, Boston, United States; [4]Regional Centre for Applied Molecular Oncology, Masaryk Memorial Cancer Institute, Brno, Czech Republic; [5]MRC-University of Glasgow Centre for Virus Research, University of Glasgow, Glasgow, United Kingdom; [6]Immunology Division, Department of Pathology, University of Cambridge, Cambridge, United Kingdom

*For correspondence:
Wilkinsongw1@cardiff.ac.uk

†These authors contributed
equally to this work
‡These authors also contributed
equally to this work

Present address: §Cardiff
School of Health Sciences,
Cardiff Metropolitan University,
Cardiff, United Kingdom

Competing interests: The
authors declare that no
competing interests exist.

Reviewing editor: Wayne M
Yokoyama, Howard Hughes
Medical Institute, Washington
University School of Medicine,
United States

**Abstract** The human cytomegalovirus (HCMV) US12 family consists of ten sequentially arranged genes (US12-21) with poorly characterized function. We now identify novel natural killer (NK) cell evasion functions for four members: US12, US14, US18 and US20. Using a systematic multiplexed proteomics approach to quantify ~1300 cell surface and ~7200 whole cell proteins, we demonstrate that the US12 family selectively targets plasma membrane proteins and plays key roles in regulating NK ligands, adhesion molecules and cytokine receptors. US18 and US20 work in concert to suppress cell surface expression of the critical NKp30 ligand B7-H6 thus inhibiting NK cell activation. The US12 family is therefore identified as a major new hub of immune regulation.

## Introduction

At 236 kb the human cytomegalovirus (HCMV) genome is the largest of any characterized human virus and is comprised of long and short unique regions ($U_L$ and $U_S$), each flanked by inverted terminal repeats. HCMV codes for around of 170 canonical protein-coding genes with 39 herpesvirus 'core' genes concentrated in the center of the $U_L$ region (*Dolan et al., 2004*). The core genes mainly encode structural components of the virion and proteins required for virus DNA replication and have orthologues in the other human herpesviruses. The vast majority of the remaining HCMV genes are not essential for virus replication *in vitro* (*Dunn et al., 2003*) yet are replete with accessory functions, many of which have been implicated in suppressing host immune responses. Unusually, HCMV encodes 15 gene families of variable size that are often clustered on the genome (*Davison et al., 2002*; *Holzerlandt et al., 2002*; *Chee et al., 1990*; *Dolan et al., 2004*; *Davison et al., 2003*). Many of these gene families exhibit homology with cellular genes and are conserved to various extents in other primate CMVs. Consequently, these primate CMV gene families are likely to have arisen through gene capture and amplification driven by differential selective pressures in their various primate hosts over millennia (*Davison et al., 2013*, *2003*).

**eLife digest** Cytomegalovirus (CMV) is one of eight herpesviruses that can infect humans. Most people will at some point become infected with CMV, yet the virus tends only to cause serious disease in people whose immune system is not working properly. Individuals living with HIV/AIDS and organ transplant recipients (who have to take drugs that suppress their immune system to prevent the organ being rejected) are particularly vulnerable to CMV infections. Critically, the virus can cross the placenta to infect of the foetus. CMV infection in the womb can cause miscarriage, lead to severe developmental problems in babies and is a major cause of deafness.

Herpesvirus infections are for life. While the immune system cannot eliminate CMV, it does have many systems that combine to sense and control infections. Natural killer cells are known to play a critical role in detecting and destroying cells infected with CMV. The virus, in turn, has nine genes that help to protect it against natural killer cells. This includes two genes that belong to a group of similar genes called the US12 family, but it is not clear whether other members of this gene family also provide protection against natural killer cells.

Fielding *et al.* now show that at least four members of the US12 gene family help CMV to evade natural killer cells. For example, two members work together to target a human protein called B7-H6 that acts a sensor to alert natural killer cells if a particular cell is infected. However, the impact of the US12 family goes even wider. The whole family works together to control proteins that are found on the surface of human cells, and many of these proteins appear to be involved in regulating the immune response.

The findings of Fielding et al. provide an insight into how the US12 gene family works, and how CMV has evolved to escape the human immune system. New therapies to control CMV infections are urgently needed so the next challenge is to design new antiviral agents that will target CMV's defence systems.

The US12 gene family consists of 10 genes, designated US12 to US21, arranged sequentially in the $U_S$ region and transcribed in the same orientation (*Chee et al., 1990*; *Dolan et al., 2004*). The genetic arrangement of the US12 family is reminiscent of 'accordion' gene expansions, which are generated when a cellular or virus resistance function is placed under strong selective pressure (*Filée, 2013*). Such an expansion was recently exemplified experimentally using a poxvirus interferon resistance function (*Elde et al., 2012*). The US12 family encodes a series of 7-transmembrane spanning proteins with low-level homology to the cellular transmembrane bax-inhibitor one motif-containing proteins (TMBIM). While not essential for virus replication, the US12 family has been implicated in HCMV tropism, virion maturation and immune evasion (*Das and Pellett, 2007*; *Cavaletto et al., 2015*; *Bronzini et al., 2012*; *Hai et al., 2006*; *Gurczynski et al., 2014*; *Fielding et al., 2014*).

Natural Killer (NK) cells play a critical role in controlling HCMV infections, and the virus invests a substantial proportion of its coding capacity to inhibit NK cell activation (*Wilkinson et al., 2013*). We previously observed that US18 and US20 suppress cell surface expression of the NK cell-activating ligand MICA (*Fielding et al., 2014*) and posited that the synergistic action of US18 and US20 may be the vestige of an immune selective pressure that drove the original expansion of the US12 family. These data show that multiple US12 family members can co-operate to target the same cellular protein. Therefore individual functions, as identified with single gene viral mutants, may not be readily replicated by expressing these same viral genes in isolation, i.e. these viral genes may work more efficiently in the context of HCMV productive infection.

To investigate the function of *all* US12 family genes, we undertook a systematic functional analysis that showed four members were NK immunevasins. Conventional biochemical investigations on US12 family proteins are rendered problematic due to their extreme hydrophobicity. We therefore undertook multiplexed Tandem Mass Tag (TMT)-based proteomic analyses to systematically evaluate the capacity of all US12 family genes to modulate the cellular proteome, both individually and in concert. Such an approach has been enabled by our recent development of Plasma Membrane Profiling (PMP) to identify novel cell surface targets for the HCMV latency protein UL138

(*Weekes et al., 2013*) and individual viral immunevasins UL141 and US2 (*Hsu et al., 2015*). Quantitative Temporal Viromics (QTV) allowed >8000 cellular and 153 viral proteins to be tracked throughout the course of productive HCMV infection, thus building a comprehensive picture of cellular control by the virus (*Weekes et al., 2014*). Through comparative analysis of ~1300 cell surface and ~7200 cellular proteins during infection with HCMV US12 family deletion mutants, we now describe in detail how this family has a profound influence not only on NK cell recognition but other key functions that impact on immunity including cellular adhesion and cytokine signalling.

## Results

### US12 family members differentially affect NK cell activation

To determine whether the US12 family has a broader role in modulating NK cell responses, a series of HCMV US12 family deletion mutants were generated (10 single deletion mutants and the US12-21 'block' deletion). The HCMV genome was manipulated by DNA recombineering in a strain Merlin BAC that did not express RL13 and UL128 (*Stanton et al., 2010*). Viruses were generated by DNA transfection and the complete genomic sequence of the virus stocks was validated by deep sequencing (*Table 1*). In NK functional assays, the △US12-21 block deletion mutant induced substantially higher levels of NK cell degranulation compared to the parent HCMV with all four different donors tested (*Figure 1B*). Significantly increased levels of NK activation were detected in assays using deletion mutants of 5 different US12 family members: US12 (3 of 4 donors), US14 (1 of 4 donors, with a trend towards increased NK activation in the other three donors), US18 (3 of 4 donors), US20 (4 of 4 donors), US21 (4 of 4 donors) (*Figure 1B*), while three US12 family deletion mutants (US15, US16, US19) reduced the level of NK cell activation in some donors (*Figure 1B*). Although members of the US12 family are capable individually of either activating or suppressing NK cell function, the net effect of the complete US12 gene family is clearly to inhibit NK cell recognition.

### The US12 family targets plasma membrane proteins

To provide an unbiased systematic analysis of the entire US12 gene family, we employed 10-plex tandem-mass tags (TMT) with MS3/Multinotch mass spectrometry to quantify whole cell (WCL) and plasma membrane (PM) proteomes in fibroblasts infected with the panel of HCMV US12 family mutants. The proteomic analyses were performed in two parts to permit inclusion of appropriate controls. Samples analyzed in Proteomic Series 1 included mock-infected controls, the parental HCMV strain, the US12-21 block deletion and defined mutants in US18, US19 and US20 (*Figure 2*, *Figure 3*, *Figure 4A*). Mass spectrometry quantified 7215 WCL and 1281 PM proteins. The extremely dynamic modulation of the host cell proteome observed during productive HCMV

**Table 1.** HCMV constructs used in the study.

| Virus | BAC # | Accession no. | Cassette used | Modification | Previous reference |
|---|---|---|---|---|---|
| HCMV | 1111 | GU179001.1 | None | RL13⁻, UL128⁻ | *Stanton et al. (2010)* |
| △US12 | 1810 | KU221097 | GalK | US12 CDS deleted | None |
| △US13 | 1831 | KU221099 | GalK | US13 CDS deleted | None |
| △US14 | 1798 | KU221093 | GalK | US14 CDS deleted | None |
| △US15 | 1800 | KU221094 | GalK | US15 CDS deleted | None |
| △US16 | 1802 | KU221095 | GalK | US16 CDS deleted | None |
| △US17 | 1804 | KU221096 | GalK | US17 CDS deleted | None |
| △US18 | 1654 | KU221091 | RpsL-Neo-LacZ | US18 CDS deleted | *Fielding et al. (2014)* |
| △US19 | 1796 | KU221092 | RpsL-Neo-LacZ | US19 CDS deleted | None |
| △US20 | 1595 | KU221090 | SacB-AmpR-LacZ | US20 CDS deleted | *Fielding et al. (2014)* |
| △US21 | 1871 | KU221100 | RpsL-Neo-LacZ | US21 CDS deleted | None |
| △US12-21 | 1815 | KU221098 | RpsL-Neo-LacZ | US12-21 deleted | None |

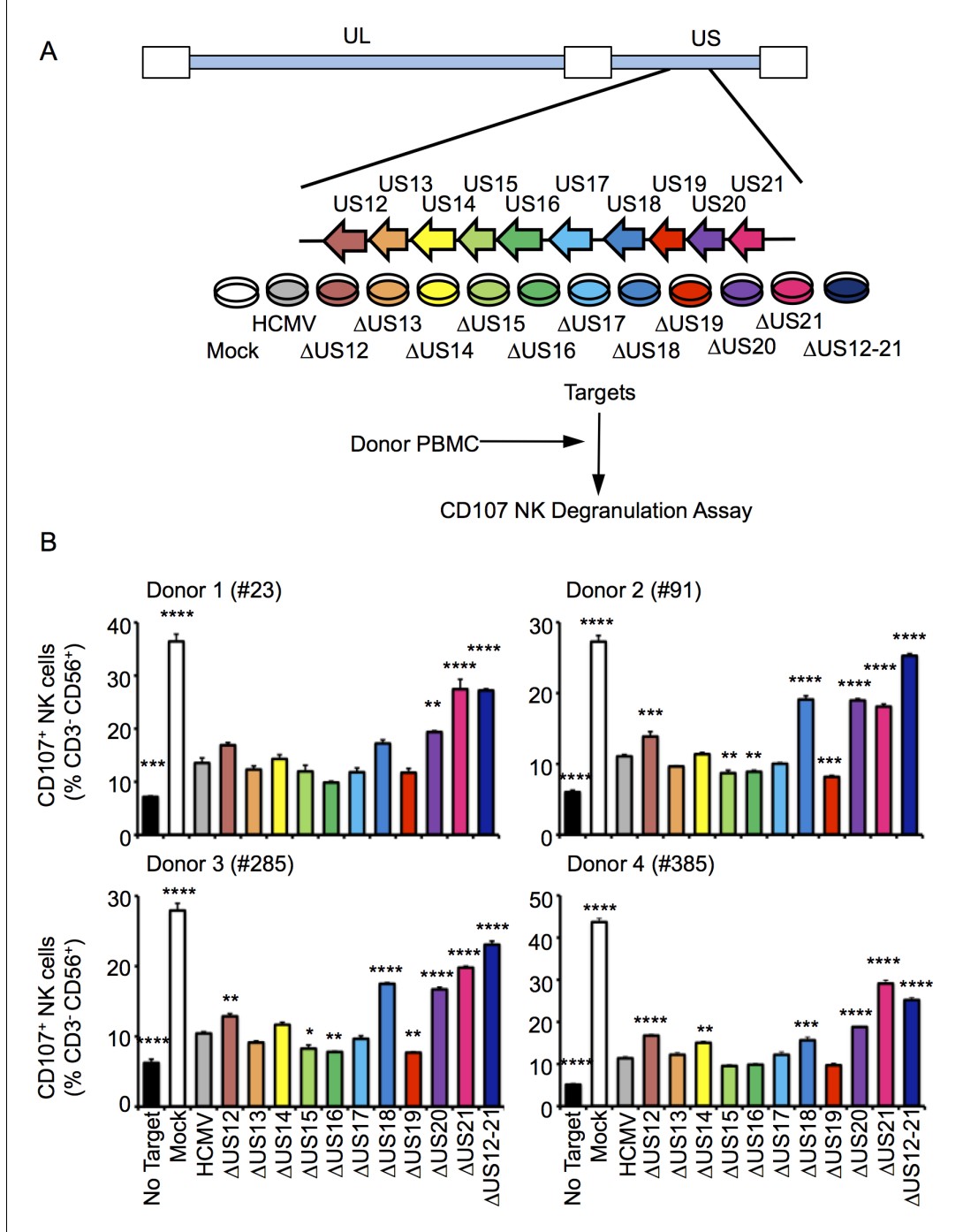

**Figure 1.** Multiple US12 family proteins regulate NK activation. (**A**) Fibroblasts (HF-TERTs) were mock infected or infected with the Merlin strain of HCMV or the series of US12 family deletion mutants for 72 hr. Infected cells were incubated with donor PBMC for 5 hr and NK degranulation assessed by % CD107$^+$ cells within the CD3$^-$, CD56$^+$ population by flow cytometry. (**B**) CD107 assay results (mean and SD) are shown from four separate donors performed in duplicate or triplicate and analysed by unpaired ordinary one way ANOVA with Dunnett's test for multiple comparisons against the HCMV control *p<0.05, **p<0.01, ***p<0.005 ***p<0.001). Infected cells were assessed by the % cells with down-regulated MHC I compared to the mock-infected cells (for the experiment using donor #23 and # 385, HCMV 93%, ΔUS12 94%, ΔUS13 98%, ΔUS14 94%, ΔUS15 98%, ΔUS16 99%, ΔUS17 98%, ΔUS18 97%, ΔUS19 99%, ΔUS20 97%, ΔUS21 91%, ΔUS12-21 94%; for the experiment using donor #91 and # 285, HCMV 94%, ΔUS12 94%, ΔUS13 94%, ΔUS14 83%, ΔUS15 95%, ΔUS16 97%, ΔUS17 85%, ΔUS18 94%, ΔUS19 96%, ΔUS20 94%, ΔUS21 94%, ΔUS12-21 96%).

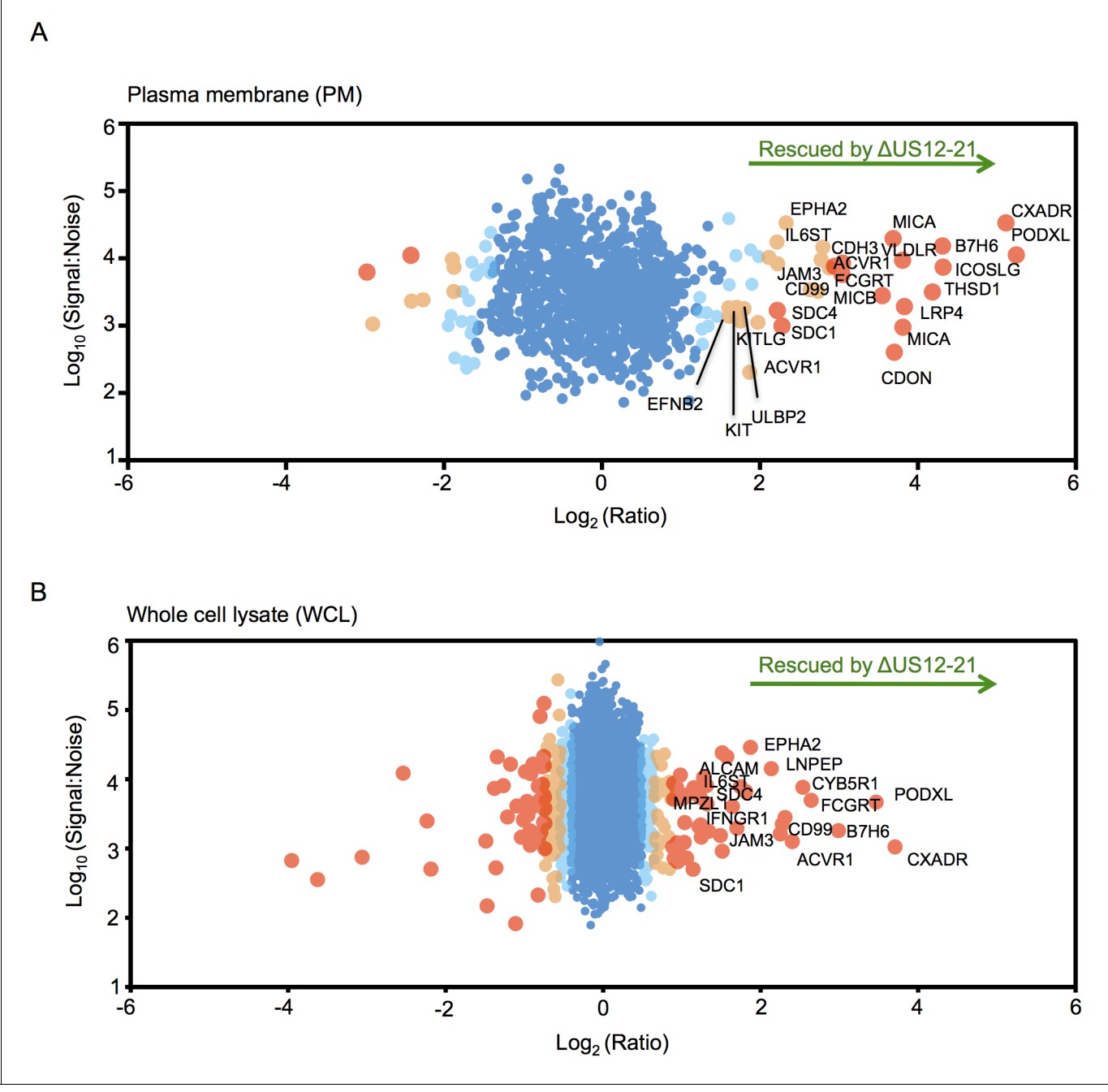

**Figure 2.** The US12 family targets numerous plasma membrane proteins. Cells infected with HCMV (Merlin) or HCMV △US12-21 mutant were processed to give PM or WCL fractions and analyzed by TMT mass spectrometry. Scatter plot of proteins identified in the PM (panel **A**) or WCL fractions (panel **B**) respectively and quantified by 2 or more unique peptides. Fold change (△US12-21-infected fibroblasts/HCMV-infected cells) is shown as the $\log_2$ ratio on the x-axis and the signal:noise on the y-axis as $\log_{10}$. Proteins unaltered by the US21-21 deletion locate at the center of the plots (0 $\log_2$/1 fold-change), whereas proteins to the left or right of center represent proteins down- or up-regulated by the US12-21 deletion respectively. Significance B was used to estimate *p* values (Cox et al., 2009). The 2 different alleles of MICA present in HFs were detected by this analysis.

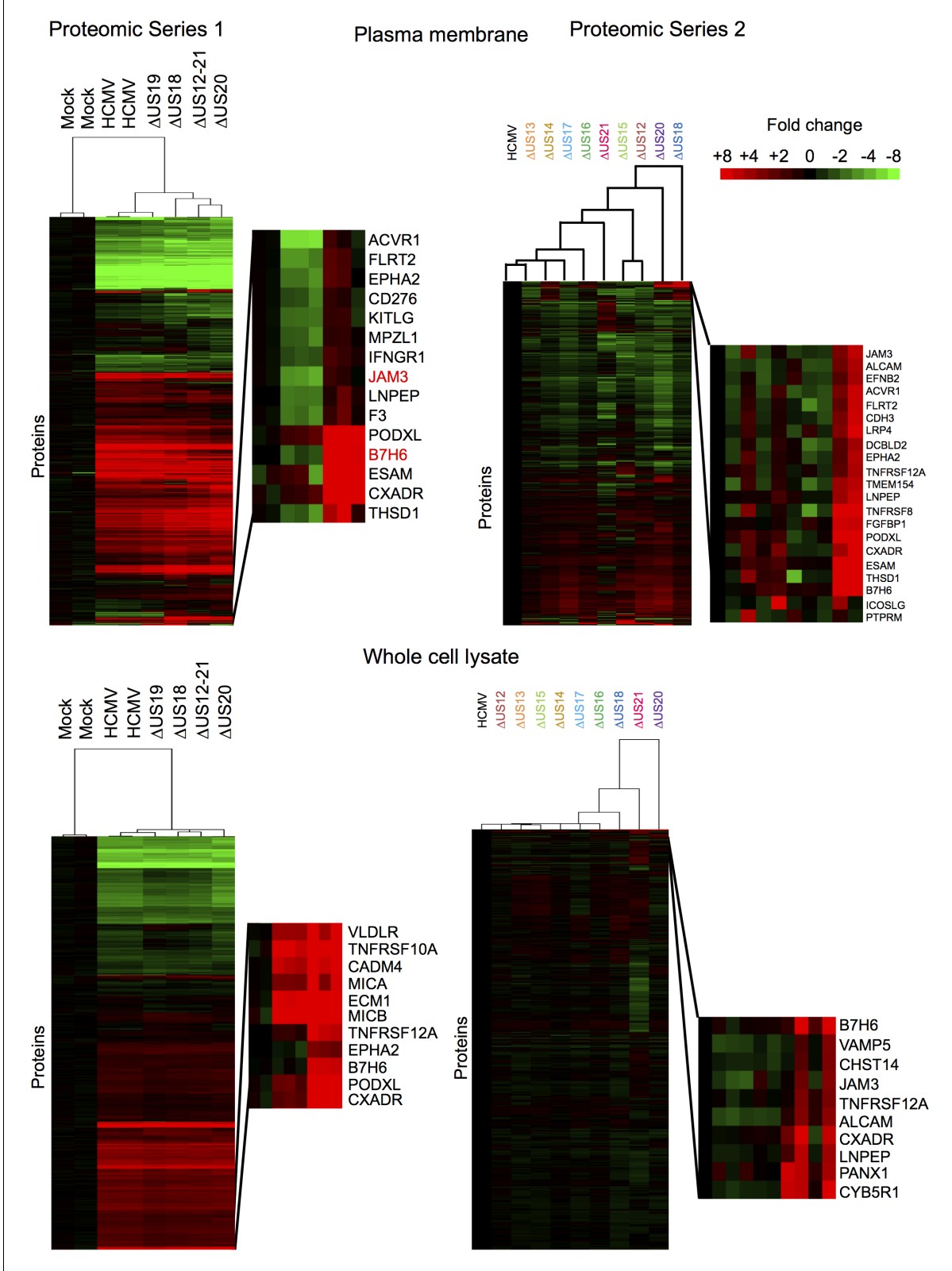

**Figure 3.** Hierarchical cluster analysis of all proteins quantified in proteomic series 1 and 2. Zoomed regions are shown for clusters of interest.

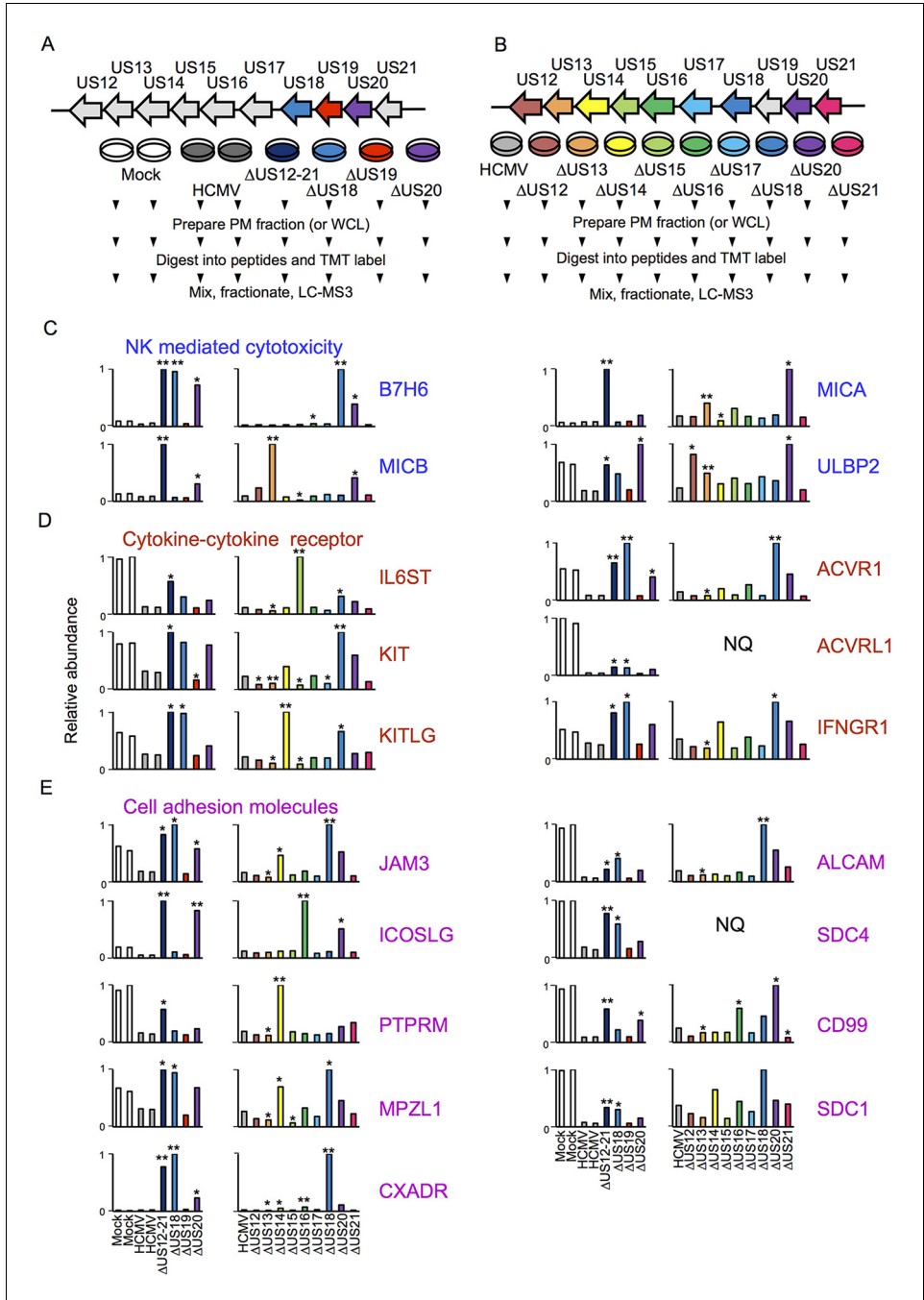

**Figure 4.** Proteomic analysis of cellular proteins targeted by US12 family members in PM samples. (**A-B**) Workflows of proteomic series 1 and 2 respectively. (**C-E**) Quantitation of PM proteins in enriched KEGG pathways identified using DAVID. Relative abundance of each protein is expressed relative to the sample with the highest abundance (set to 1). NQ – not quantified. A number of the US12-21 mutant targets were also regulated by the US18 and US20 mutants (B7-H6, ULBP2, IL6ST, KIT, KITLG JAM3, ACVR1, ACVRL1, IFNGR1, JAM3, MPZL1, CXADR, ALCAM, SDC4, CD99, SDC1). A subset of these proteins were also regulated by the US14 and/or US16 mutants (KIT, KITLG, ACVR1, IFNGR1, JAM3, MPZL1, CD99, SDC1). We estimated p values for the ratios of each mutant compared to HCMV Merlin using Benjamini-Hochberg corrected Significance B values (*Cox and Mann, 2008*): *p<0.05, **p<0.0001. For proteomic series 1, ratios were calculated as US12 family deletion mutant / average HCMV and for proteomic series 2, US12 family deletion mutant / HCMV. All proteins quantified by 2 or more peptides were included in this calculation. SDC1 was quantified by a single peptide in proteomic series 2.

infection was consistent with previous findings (*Figure 3*)(*Weekes et al., 2014*). The role of the entire US12 gene family was assessed by comparison of the HCMV US12-21 block deletion mutant with its parental virus. The impact was most noticeably focused on the PM, in that the majority of WCL proteins affected were also PM 'hits' (*Figure 2*). The heat map illustrating changes in protein abundance correspondingly appears more dynamic i.e. a higher proportion of proteins detected were regulated in response to the deletion of individual US12 family genes for the PM proteome than for the much larger set of proteins quantified in WCL samples (*Figure 3*). For a given PM protein modulated by the US12 gene family (*Figure 4*), comparable results were observed in WCL samples (*Figure 5*), suggesting that the US12 family regulates protein expression or stability of cell surface proteins.

NK cells continually monitor the levels of inhibitory and activating ligands on the surface of potential targets, thus one of the roles of the US12 family in re-modeling the PM proteome is compatible with its role in directly impacting NK cell recognition. To gain an overview of pathways targeted, the online bioinformatics resource DAVID (Database for Annotation, Visualisation and Intergrated Discovery; https://david.ncifcrf.gov/) was used to perform a KEGG (Kyoto Encyclopedia of Genes and Genomes) pathway enrichment analysis on proteins rescued >3 fold by the deletion of the US12-21 block (*Dennis et al., 2003*; *Huang et al., 2009*). Multiple KEGG pathways were significantly enriched (data not shown), including natural killer cell-mediated cytotoxicity (MICA, MICB, ULBP2, IFNGR1), cytokine-cytokine receptor interaction (IL6ST, KIT, KITLG, ACVRL1, ACVR1, IFNGR1) and cell adhesion molecules (CAMs; JAM3, ICOSLG, PTPRM, MPZL1, CXADR, ALCAM, SDC4, CD99, SDC1). The modulation of cell adhesion molecules and chemokines/cytokines may additionally impact NK cell recognition. A further NK ligand, B7-H6, was not identified by DAVID analysis, but was also regulated by the US12-21 mutant. While MICA is a recognized target of US18 and US20 (*Fielding et al., 2014*), the regulation of the NK cell activating ligands B7H6, MICB and ULBP2 by the US12 family are novel.

## Functional independence and co-operation exerted by family members

A substantial subset of the proteins regulated by the US12-21 block mutant were similarly modulated by US18 and US20 (*Figure 4*, *Figure 5*), whereas US19 specifically targeted RALGPS2 (*Figure 6*). To determine the contribution of each individual US12 family member to the overall effects observed with the 'block' deletion mutant, Proteomic Series two compared infection with all the single gene deletion mutants (except the US19 deletion) to the parent HCMV (*Figure 4B*). Mass spectrometry quantified 7156 whole cell and 1312 PM proteins. We re-examined key candidate molecules identified by bioinformatic analysis of the US12-21 block deletion (*Figure 4C–E*). A striking feature of this data was that the majority of these target proteins were regulated by both US18 and US20, with a subset additionally regulated to a lesser degree by US14 and/or US16 (*Figure 4C–E*). Other US12 family members also had important effects, with US12 and US13 mutants regulating the NKG2D ligands (NKG2DL) ULBP2 and MICB respectively, US14 mutant regulating PTPRM, and the US15 mutant regulating IL6ST (*Figure 4C–E* and *Figure 6*). Overall, the contribution made by a given family member varied dramatically ranging from the highly focused impact of US13 and US19 (one cellular target each) to the exceptionally promiscuous US20 (54 cellular targets) (*Figure 6*). Where a given protein was quantified both in PM and WCL samples, generally similar changes were observed in both samples (*Figure 4* and *5*). Interestingly, MICA was regulated by the deletion of US12-21 block to a much greater extent than any of the single deletion mutants, implying that multiple US12 family members may need to act in concert to optimize control over certain cellular targets (*Figure 4C*).

Infection with the US21 deletion mutant resulted in selective impairment of many late phase Tp3 and Tp5 (Temporal protein profile 3 and 5) HCMV proteins (*Weekes et al., 2014*), for example UL32 and UL99/pp28, and elevated expression of pUS20, whereas the HCMV US12-21 block mutant did not recapitulate this effect (*Figure 7*, with further proteomic data searchable in Excel spreadsheet *Supplemental file 1*). Our data are consistent with deletion of the US21 structural gene impacting on the transcriptional control of US20 (data not shown), and may contribute to a potential growth defect observed for this particular mutant. Therefore the activation of NK cells in response to US21 mutant-infected targets cannot be assigned directly to an effect of US21 (*Figure 1*). A definitive assessment of the function of US21 will require an alternative approach.

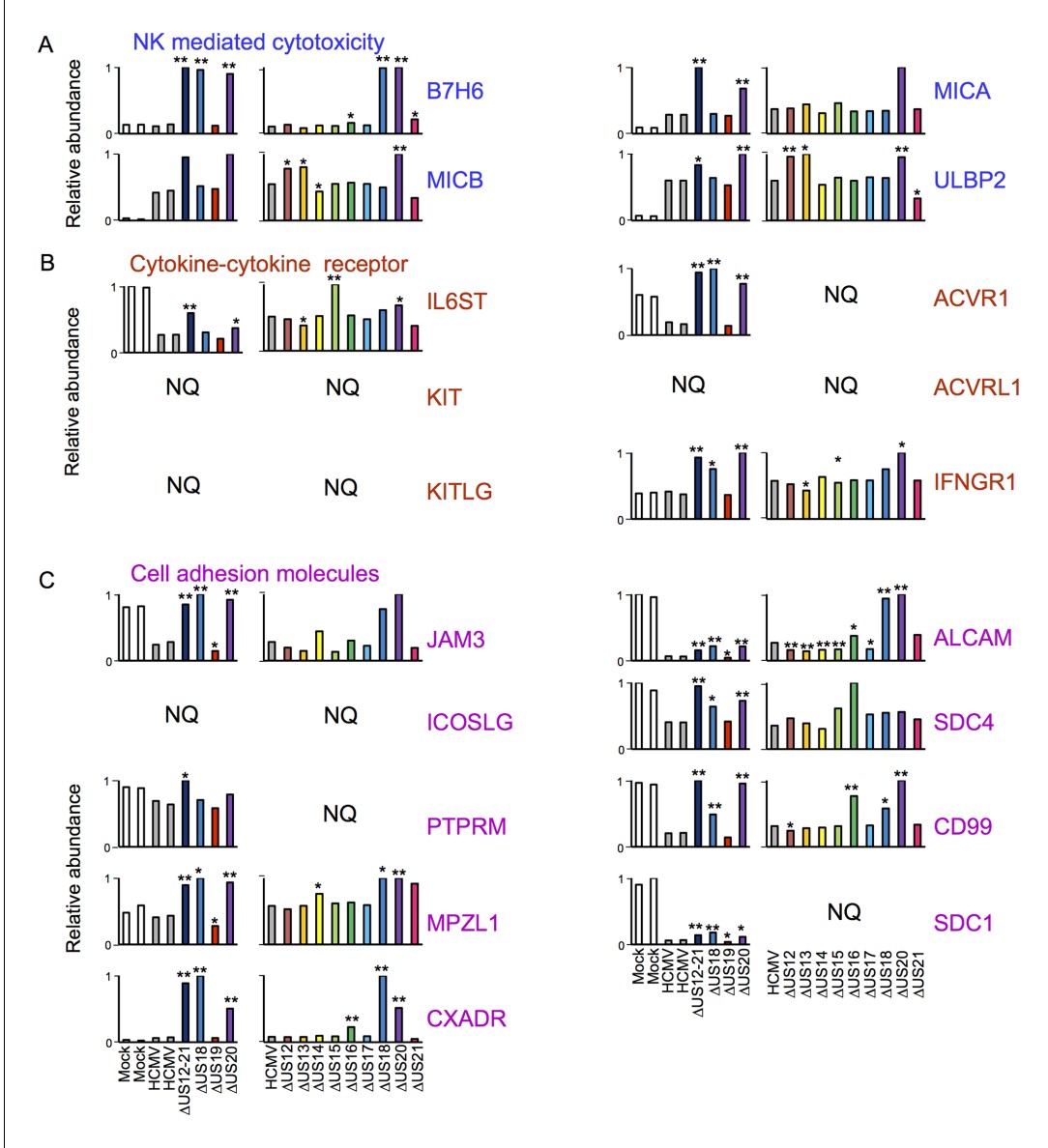

**Figure 5.** Individual US12-family proteins target key natural killer cell ligands, cell adhesion molecules and cytokines and their receptors in WCL samples. (**A–C**) Quantitation of key US12 family targets in WCL samples (proteomic series 1 and 2 - comparative analysis of these proteins from PM samples is shown in *Figure 3*). We generally observed similar results for proteins quantified in both PM and WCL samples. Relative abundance of each protein is expressed relative to the sample with the highest abundance (set to 1). NQ – not quantified. P values were calculated as described in *Figure 4*: *p<0.05, **p<0.0001. MICB was only quantified by 1 peptide in proteomic series 1, and MICA, JAM3 and SDC4 by one peptide in proteomic series 2.

Of all PM proteins regulated >3 fold by a member of the US12-21 family, 29% were regulated by 2 or more family members and 6% by 3 or more family members (*Figure 6*), although using a stringent 3-fold cutoff may underestimate the true incidence of co-regulation. Such co-regulation may suggest evolutionary pressure underpins the expansion of this gene family.

## Effect of lysosomal inhibition on US12 family targets

The majority of proteins that the US12 family down regulates from the PM are also lost from the WCL, consistent with post-translational proteolysis or an overall reduction in expression. We previously found that US20 targets the NKG2DL MICA for degradation in lysosomes (*Fielding et al.,*

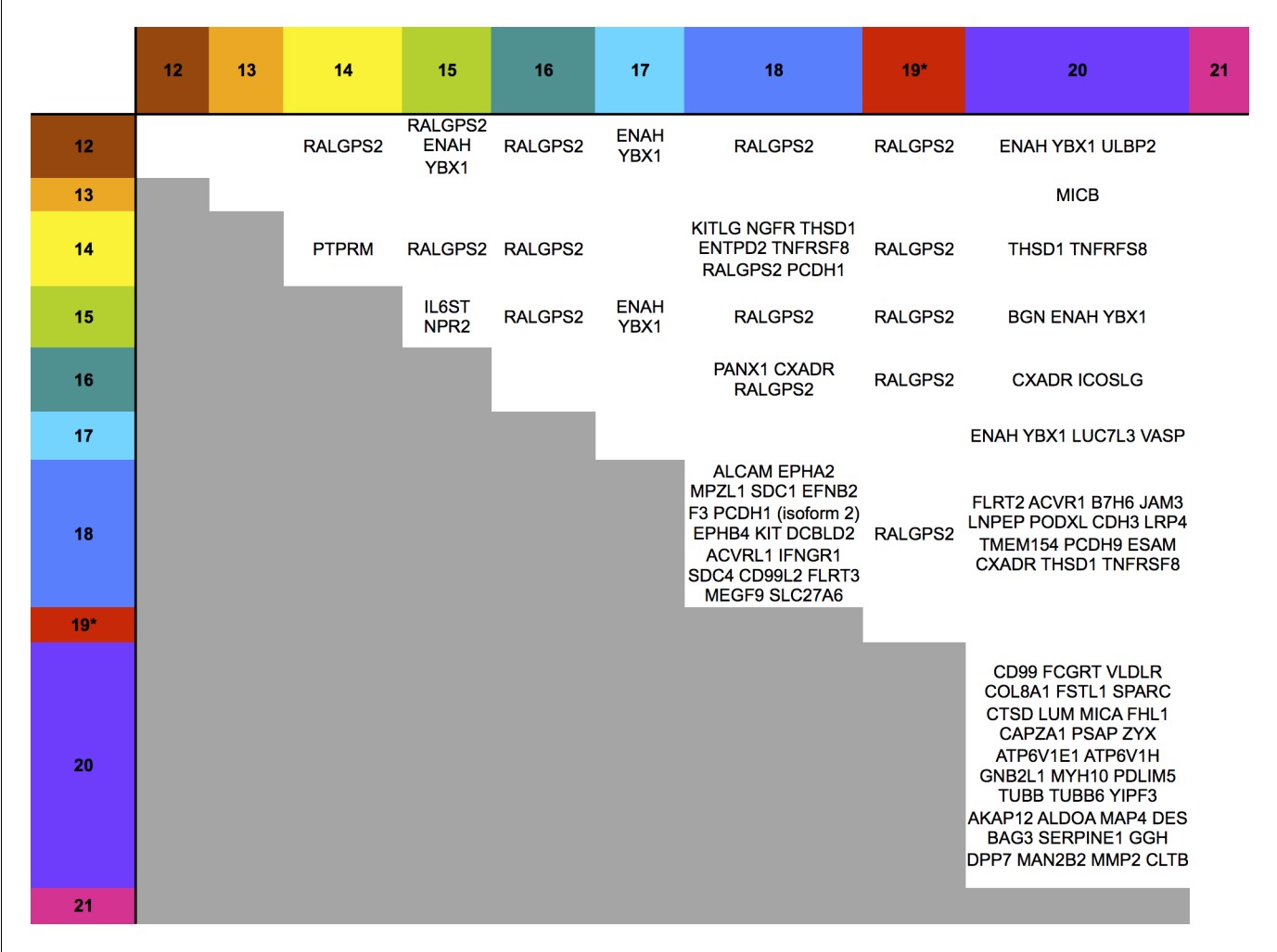

**Figure 6.** Co-regulation of multiple cell surface proteins by >1 US12 family gene. Proteins that were rescued by deletion of each US12 family member in the PM fraction of proteomic Series 2 (i.e. not including US19) were analyzed to determine which were additionally targeted by one or more US12 family members. Proteins were plotted in a matrix with unique protein targets on the lower diagonal of the plot (US12 family member compared with itself) and common protein targets in intersections with other US12 members. To identify the highest confidence targets of the US12-21 family for the purposes of this analysis and to generate a shortlist of US12-21 family PM 'hits', we employed the following strategy: we included proteins (a) exhibiting at least 3-fold rescue upon deletion of a given US12 family member, quantified by at least two peptides and annotated by GO to indicate a PM location. (b) validated by a corresponding >2 fold change in at least one of (i) deletion of the US12-21 block (proteomic series 1 or 3), (ii) the biological repeat of US18 or US20 in proteomic series 1, (iii) the corresponding gene deletion in WCL proteomic series 2. This strategy identified all proteins shown in *Figure 3*. *US19 data was used from proteomic series 1.

*2014*). The DAVID pathway analysis identified increases in lysosomal proteins within the PM fraction of △US20-infected cells; the list of lysosomal proteins included a number of cathepsins and ATPases (PSAP, CTSD, ATP6V1E1, ATP6V1H, NPC2) (data not shown). These data suggest a role for US20 in regulating intracellular endo-lysosomal vesicular transport.

To determine if this is a more general mechanism for the whole US12-21 family, HFFs were infected with HCMV, the US12-21 block deletion mutant or HCMV in the presence of the lysosomal protease inhibitor leupeptin for 12 hr prior to harvest, and protein expression was analyzed by 10-plex TMT at 24 hr, 48 hr and 72 hr post-infection (Proteomic Series 3, *Figure 8A*).

The △US12-21 'block' deletion caused substantial shifts in WCL proteins, with >80 proteins rescued >2 fold at 48 hr, compared to the parent viral infection (*Figure 8B*). Of these, 26% were also rescued >2 fold and 51% > 1.5 fold with leupeptin suggesting that the US12 family targets multiple

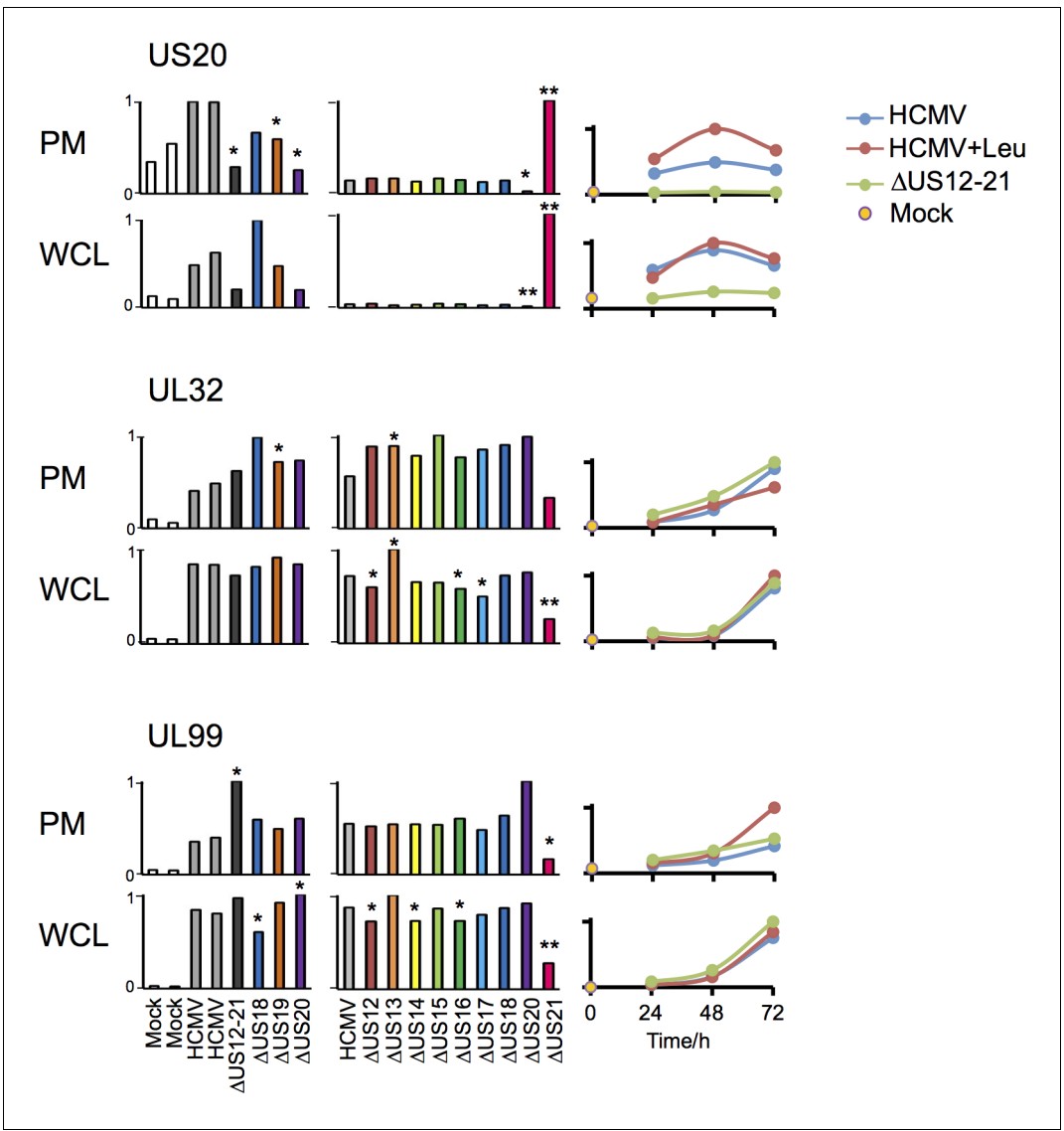

**Figure 7.** Altered expression of US20, UL32 and UL99 levels in the US21 deletion mutant-infected fibroblasts. Quantitation of US20, UL32 and UL99 in proteomic series 1–3 in both PM and WCL. Abundance of each protein is expressed relative to the sample with the highest abundance (set to 1). For proteomic series 1 and 2, p values were calculated as described in *Figure 4*: *p<0.05, **p<0.0001. US20 was only quantified by one peptide in proteomic series 1, WCL experiment.

host proteins to the lysosome. Leupeptin treatment had a more limited effect on rescuing PM proteins back to the the cell surface (*Figure 8B*), consistent with our previous observation of intracellular accumulation of MICA without protein relocalisation to the cell surface during treatment with leupeptin or other lysosomal inhibitors (*Fielding et al., 2014*). Of the 21 proteins identified in *Figure 4* within the key categories 'natural killer cell-mediated cytotoxicity', 'cytokine-cytokine receptor interaction' and 'cell adhesion molecules', 11 were rescued >2 fold by leupeptin, with the remainder exhibiting 1.3–2-fold rescue (*Figure 8C*). MICB was clearly rescued by leupeptin treatment (*Figure 8C*). Interestingly, control of MICB and ULBP2 levels correlated with regulation of UL16 by US13 and US12 (*Figure 9*). However, leupeptin rescue of ULBP2 and UL16 was less convincing than for MICB (*Figure 9*). Although many cellular substrates of the US12 family are clearly being targeted for lysosomal degradation, there may be additional mechanisms by which this family regulates the PM proteome.

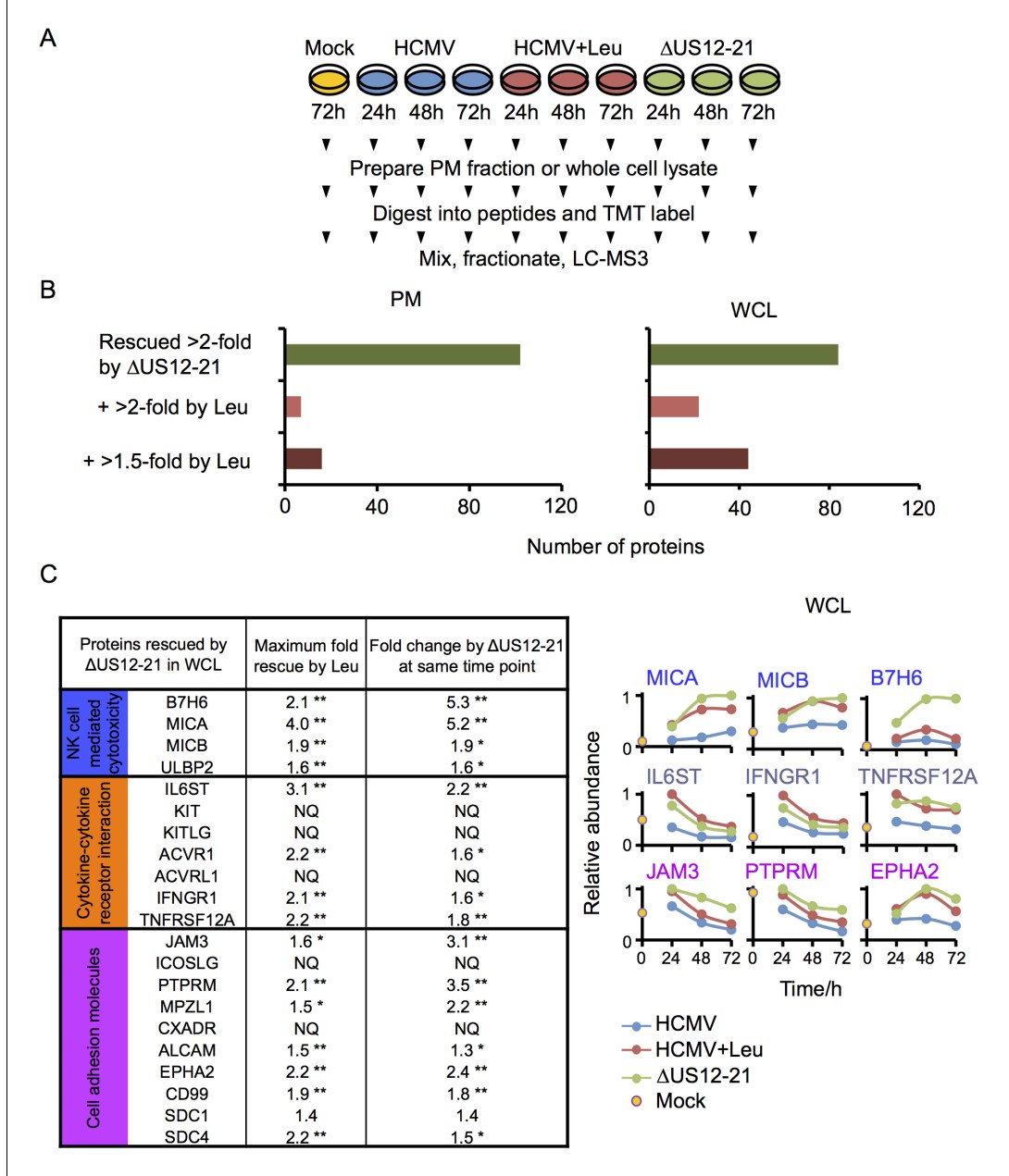

**Figure 8.** Multiple US12 gene family targets are degraded via the lysosomal pathway. (**A**) Workflow of Proteomic Series 3. (**B**) Number of proteins targeted by the US12-21 block and additionally rescued >2 fold in the dataset by the △US12-21 deletion mutant or rescued >2 fold or >1.5 fold by leupeptin treatment for both WCL and PM. C. Comparable degree of rescue of US12 target proteins by US12 deletion and leupeptin treatment. Quantitation of a subset of these proteins is shown relative to the maximum abundance (set to 1). p-values were calculated as described in *Figure 4*, for the ratios of leupeptin treatment or US12-21 deletion virus infection at each time point compared to the matched wild-type Merlin-infected control. *p<0.05, **p<0.0001. SDC1 was only quantified by one peptide.

## Validation of PM protein regulation by US12 family members

We used flow cytometry to validate a proportion of the various PM protein targets that are regulated by the US12 family (*Figure 10*). Next, we examined NKG2DL MICA and MICB (*Figure 10*). We found that MICA expression was rescued to a greater degree with the US12-21 block mutant than with the individual US18 or 20 deletion mutants (*Figure 10*). This is consistent with our previous analysis showing that a combined US18 and US20 deletion had a greater effect on cell surface MICA

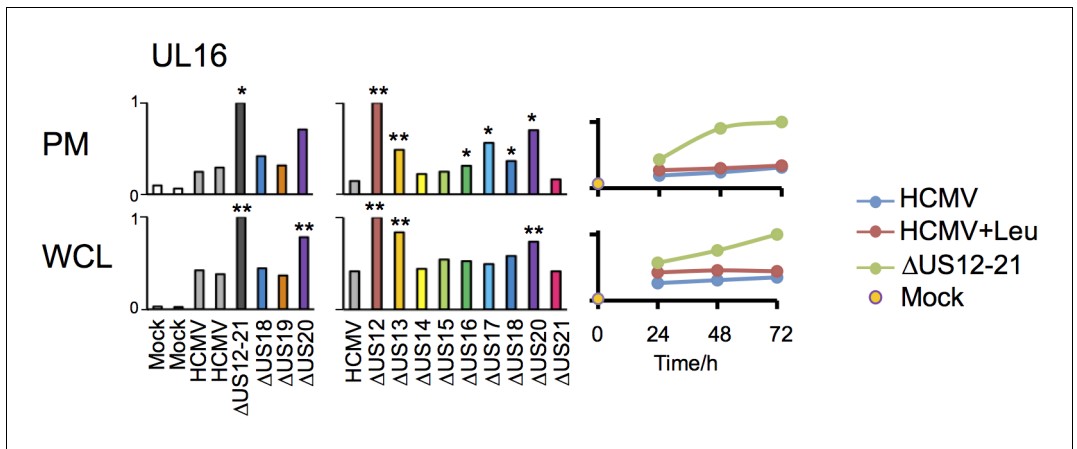

**Figure 9.** Regulation of UL16 levels by the US12 family. Quantitation of UL16 in proteomic series 1–3 in both PM and WCL. Relative abundance of each protein is expressed relative to the sample with the highest abundance (set to 1). For proteomic series 1 and 2, p values were calculated as described in *Figure 4*: *p<0.05, **p<0.0001.

than single gene deletion mutants (*Fielding et al., 2014*). The US13 deletion mutant caused a minor, but consistent, elevation in cell surface expression of MICB. Other US12 family genes may yet contribute to regulating MICB, as there was a further increase in the US12-21 block mutant (*Figures 10* and *4C*).

## B7-H6 is an NK cell activating ligand targeted by US18 and US20

The fact that, in addition to targeting MICA, US18 and US20 together are also implicated in targeting B7-H6 is extraordinary. B7-H6 is a known ligand for NKp30 and exogenous expression in fibroblasts increased NK degranulation (*Figure 11A–C*). Ectopic expression of US20, and to a lesser extent US18, from adenovirus vectors reduced levels of exogenously expressed B7-H6 detected by immunoblotting (*Figure 11D*). Both PM and WCL expression of B7-H6 was induced by the HCMV US18, US20 and US12-21 deletion mutants but not by the parental virus (*Figure 4C*, *Figure 5A*), which was confirmed by flow cytometry and immunoblot (*Figures 10* and *12A*). B7-H6 is therefore induced as a 'stress ligand' during productive infection, but its expression is controlled by US18 and US20, acting in concert, to target it for proteolysis (*Figure 8C*). Being the dominant ligand for the NK activating receptor NKp30, B7-H6 is potentially a critical target for HCMV.

We investigated the effect of the US12 family members upon activation of an NKp30-responsive reporter cell line, and found that the presence of US18 and US20 was required to inhibit reporter activity (*Figure 12B*). Reporter cell fluorescence could be inhibited either by transfection of B7-H6 specific siRNA or by use of a B7-H6 specific blocking antibody (CH31) (*Figure 12C and D* respectively). The presence of both US18 and US20 was required to prevent B7-H6 activating the NKp30 reporter function.

We sought to differentiate the functional impact exerted by US18 and US20 on B7-H6 and MICA. B7-H6 knockdown in cells infected with HCMV US18 and US20 deletion mutants was correspondingly able to reduce donor NK cell activation (*Figure 13A and B*). The impact of B7-H6 knockdown on the HCMV US12-21 block mutant was more variable and consistent with the US12 gene family encoding additional NK modulating functions (*Figure 13B*, *Figure 1*). A B7-H6 blocking antibody had an even more pronounced effect, reducing NK activation in response to the US18, US20 and US12-21 deletion mutants to levels of the parental HCMV control (*Figure 13C*). These results are consistent with B7-H6 playing a major role in NK cell recognition of HCMV-infected cells, a function that is countered by the US12 family members US18 and US20.

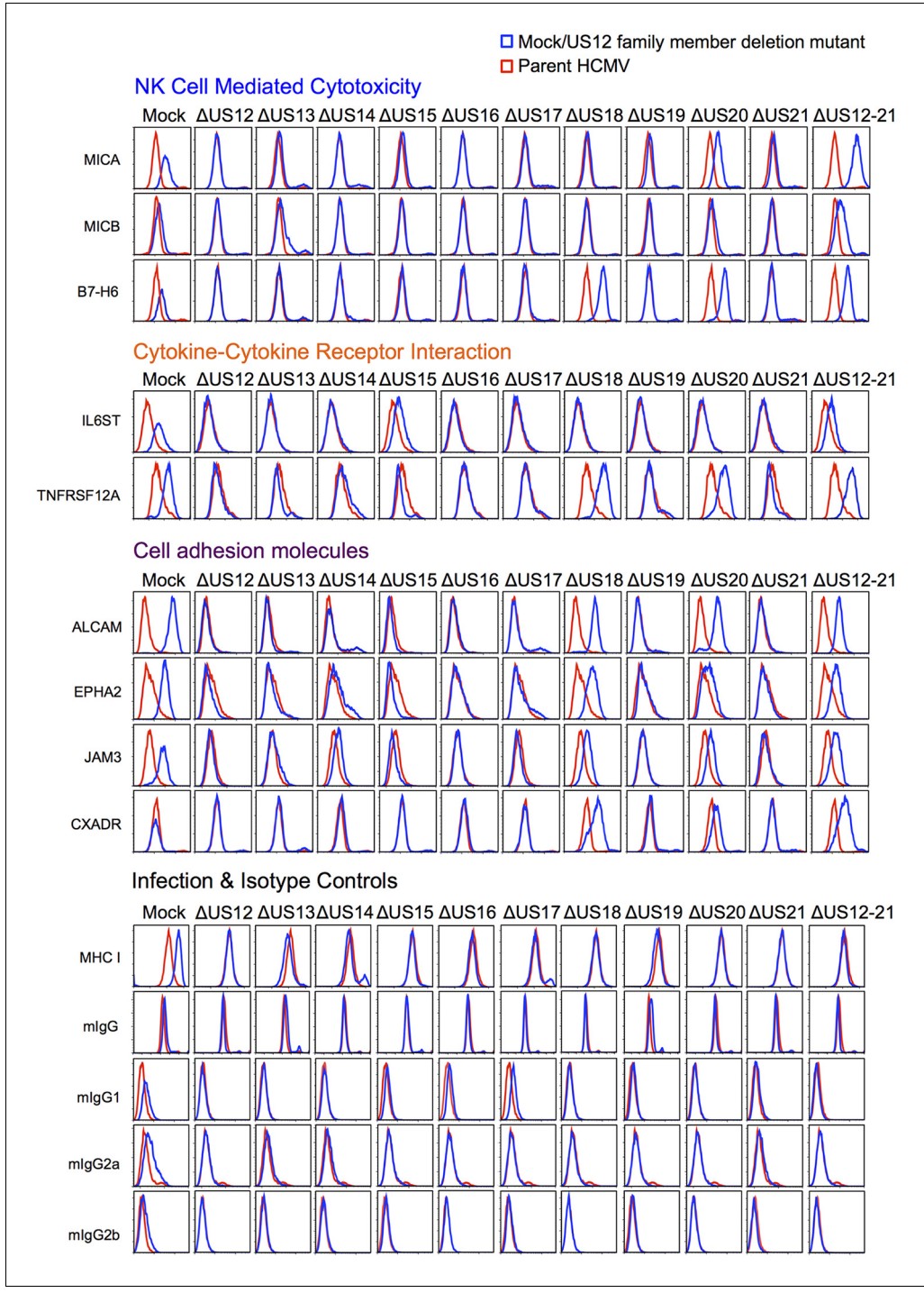

**Figure 10.** Validation of cell surface proteins regulated by the US12 family. Flow cytometry confirmed proteomic data for proteins representative of each category enriched in the DAVID analysis. Staining in mock/US12 family member deletion mutant infections (blue line) is shown relative to the parental HCMV infection (red line). Flow cytometry was carried out for cell surface expression of MHC I (W6/32) as a control for HCMC infection and isotype antibody staining controls (with directly PE-conjugated IgG1, IgG2a or IgG2b antibodies or for unconjugated antibodies mIgG and an anti-mouse-AF647 conjugated secondary antibody). Infected cells were assessed by the % cells with down-regulated MHC I compared to the mock-infected cells (HCMV 94%, ΔUS12 94%, ΔUS13 94%, ΔUS14 83%, ΔUS15 95%, ΔUS16 97%, ΔUS17 85%, ΔUS18 94%, ΔUS19 96%, ΔUS20 94%, ΔUS21 94%, ΔUS12-21 96%). Results are representative of at least two independent experiments.

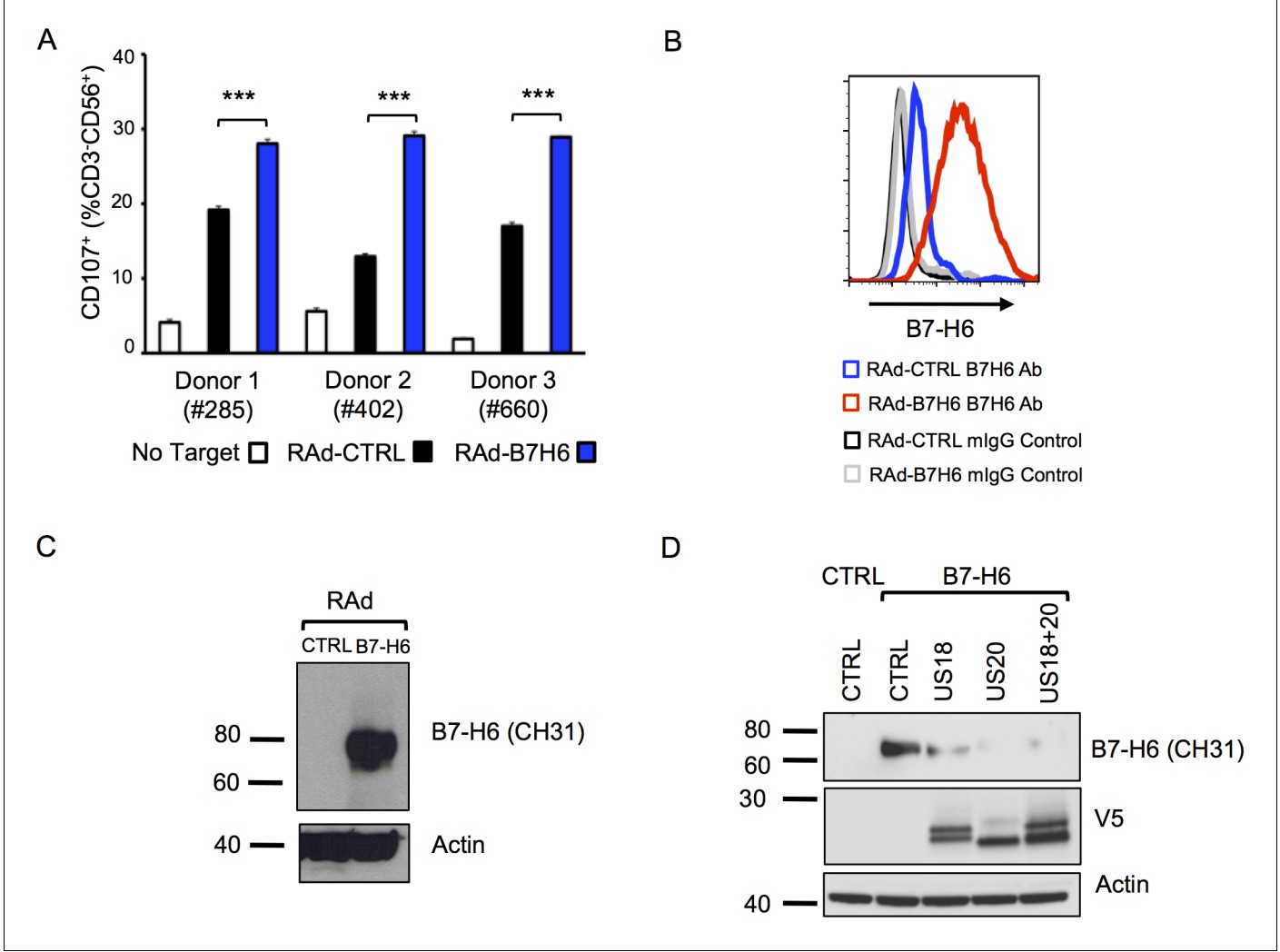

**Figure 11.** Adenovirus expressed B7-H6 regulates NK cell activation and B7-H6 levels are regulated by ectopically-expressed US18 and US20. (A) HF-CARs were infected with control (RAd-CTRL) or B7-H6-expressing (RAd-B7-H6) adenovirus vectors (MOI 5). Cells were harvested 48 h p.i. and used as targets in a CD107 degranulation assay with buffy-coat derived PBMC from 3 separate donors in duplicate or triplicate. Results (shown as mean and SD) were analyzed by unpaired two-tailed Student's T-test. ***p<0.001, ****p<0.0001. (B and C) HF-CARs were infected with control (RAd-CTRL) or B7-H6-expressing (RAd-B7-H6) adenovirus vectors (MOI 5). Cells were harvested 48 h p.i. and B7-H6 cell surface expression analyzed by flow cytometry (B) or western blot (C). Results are representative of at least 2 independent experiments. (D) HF-CARs were infected with adenovirus control (CTRL), US18-expressing adenovirus, US20-expressing adenovirus or a combination of both US18 and US20 (MOI 5 each, made up to a total MOI of 10 with RAd-CTRL). After 24 hr incubation, HF-CARs were infected with adenovirus control (CTRL) or B7-H6-expressing adenovirus (B7–H6) at MOI 5 as indicated before incubation for a further 48 hr. Whole cell lysates were prepared and analysed by immunoblotting with the antibodies indicated. Results are representative of two independent experiments.

## Discussion

The HCMV genome contains 15 gene families of various sizes that have been acquired during evolution to promote virus persistence. Clusters of US12-related genes can be detected in cytomegaloviruses of New World primates, i.e. Green Monkey and Owl Monkey CMVs, thus the capture and expansion of an ancestral precursor presumably took place over 41 million years ago (*Davison et al., 2013*). Within Cynomolgus and Rhesus macaques (Old World primates), chimpanzee and human CMVs, the US12 family is maintained as a well conserved contiguous tandem array of 10–11 genes located in the same relative position on each genome. Amongst circulating HCMV clinical strains,

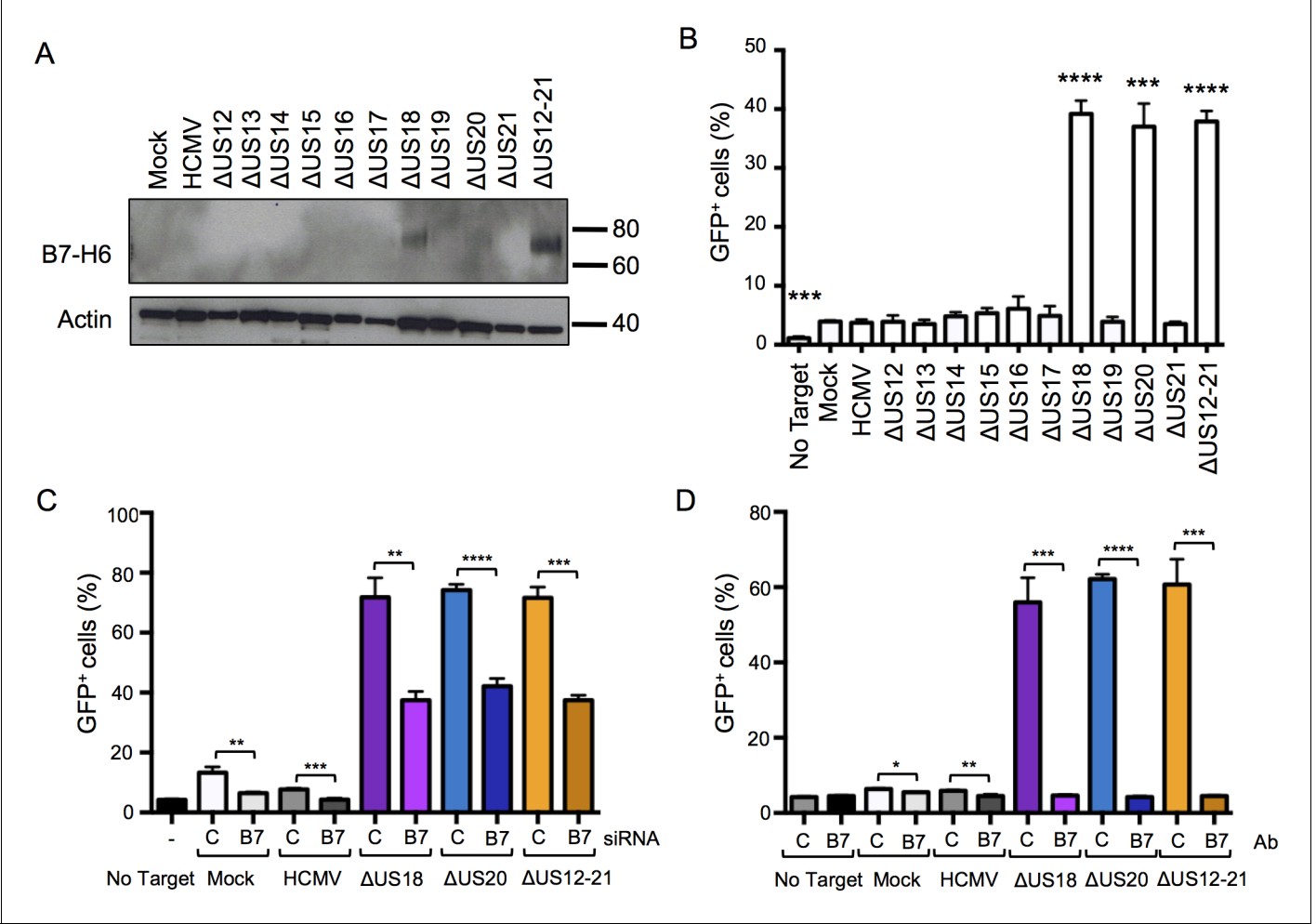

**Figure 12.** Differences in B7-H6 levels on HCMV US12 family deletion mutant-infected cells regulate NKp30-mediated responses. (A) Whole cell lysates were prepared from mock, HCMV or the series of US12 family deletion mutant infected fibroblasts and analyzed by immunoblotting with antibodies specific for B7-H6 (CH31) or actin. Results are representative of two independent experiments. (B) An NKp30-responsive 2B4 reporter cell line containing a NFAT-GFP reporter construct (CT299) was incubated for 24 hr with mock, HCMV or the series of US12 family deletion mutant infected fibroblasts in triplicate. Fixed cells were then analyzed for GFP fluorescence by flow cytometry compared to cells incubated with no target fibroblasts. Results are expressed as the % GFP positive reporter cells (mean and SD) and were analyzed by unpaired two-tailed Student's T-test. Results are representative of two independent experiments. (C) HF-TERTs were transfected with control (C) or B7-H6 (B7) siRNAs for 24 hr prior to infection with the parent HCMV, △US18, △US20 or △US12-21 mutants for 72 hr. Cells were then incubated for 24 hr with CT299 NKp30 reporter cells in triplicate and GFP+ cells determined by flow cytometry, and analysed by unpaired two-tailed Student's t-test. Results (mean and SD) are representative of two independent experiments. (D) HF-TERTs infected with HCMV, △US18, △US20 or △US12-21 mutants were incubated for 24 hr with CT299 NKp30 reporter cells, in the presence isotype control (C) or B7-H6 blocking antibodies (B7) in triplicate, GFP+ cells determined by flow cytometry and analyzed by unpaired two-tailed Student's t-test. Results (mean and SD) are representative of two independent experiments.

the US12 family exhibits relatively high levels of sequence conservation and genetic integrity (*Sijmons et al., 2015*).

The complexity of the HCMV genome, its restricted host range in vitro and protracted replication cycle has frustrated studies into HCMV gene function. However, the advent of high-resolution multi-plexed proteomics is revolutionizing our understanding of how HCMV orchestrates host cell gene expression and evades host defenses (*Weekes et al., 2014*; *Hsu et al., 2015*). By systematically ana-lyzing a bespoke panel of HCMV deletion mutants, we have discovered that the US12 family selec-tively targets a broad range of plasma membrane proteins that include, not only NK cell activating ligands, but also T-cell co-stimulatory molecules, cell adhesion molecules, and cytokine/cytokine receptors. The diversity of proteins targeted by many of the US12 family implies a broad strategy of

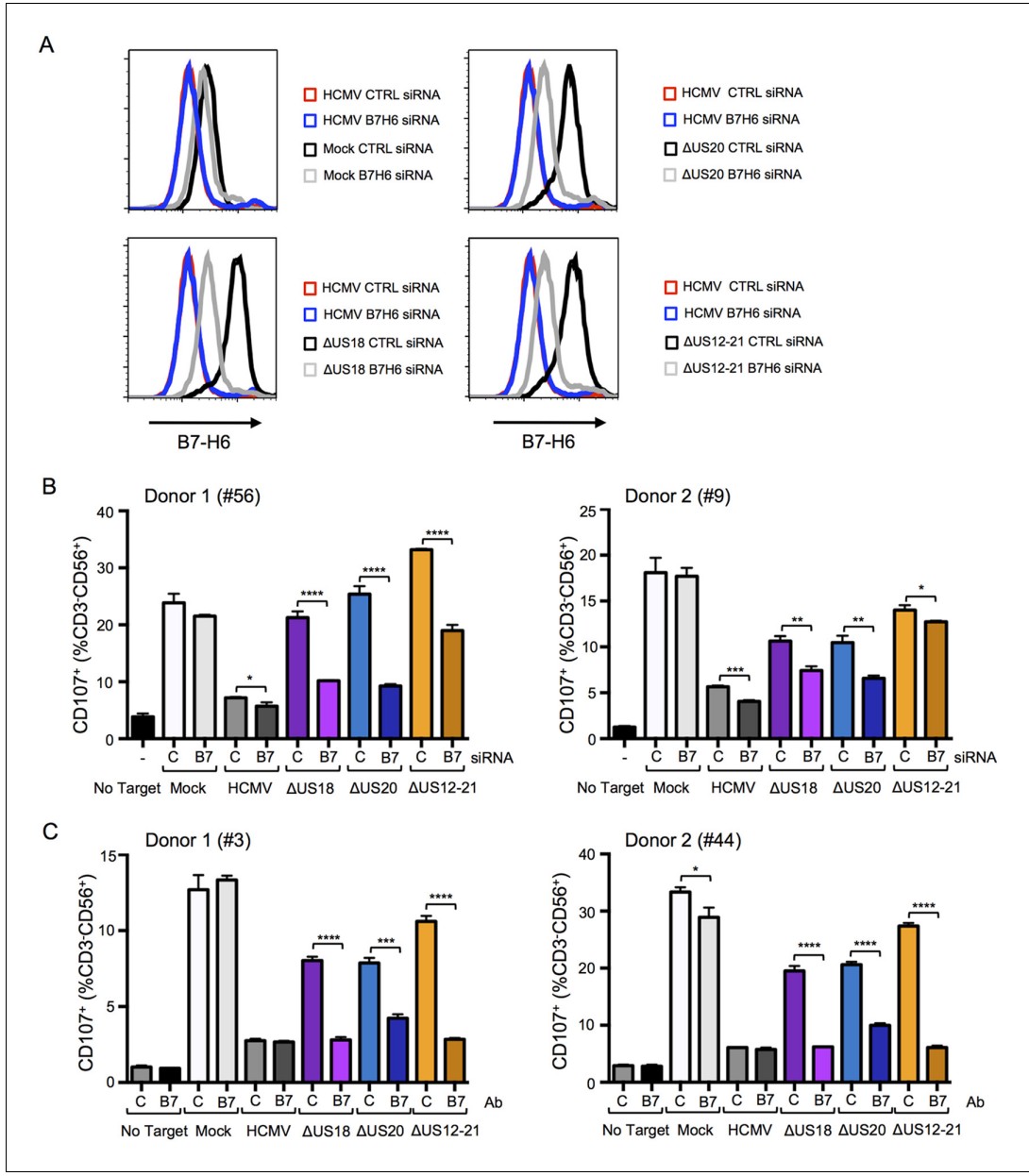

**Figure 13.** B7-H6 regulation has a major effect on NK activation in response to HCMV infection. (**A** and **B**) HF-TERTs were transfected with control (**C**) or B7-H6 (**B7**) siRNAs for 24 hr prior to infection with the parent HCMV, △US18, △US20 or △US12-21 mutants. Cells were harvested 72 h p.i. and B7-H6 cell surface expression analyzed by flow cytometry (**E**) or used as targets in a CD107 degranulation assay with donor-derived PBMC in triplicate (**F**). Results (mean and SD) were analyzed by unpaired two-tailed Student's t-test and are representative of two independent experiments using 2 separate donors in each. Infected cells were assessed by the % cells with down-regulated MHC I compared to the mock-infected cells (HCMV CTRL siRNA 95%, △US18 CTRL siRNA 91%, △US20 CTRL siRNA 93%, △US12-21 CTRL siRNA 86%, HCMV B7-H6 siRNA 96%, △US18 B7-H6 siRNA 84%, △US20 B7-H6 siRNA 96%, △US12-21 B7-H6 siRNA 79%). (**C**) HF-TERTs infected with HCMV, △US18, △US20 or △US12-21 mutants were used as targets in a CD107 degranulation assay with donor-derived PBMC, in the presence isotype control (**C**) or B7-H6 (**B7**) blocking antibodies in triplicate. Results (mean and SD) were analyzed by unpaired two-tailed Student's t-test and are shown for two separate donors. *p<0.05, **p<0.01, ***p<0.001, ****p<0.0001.

redirecting cell surface receptors. A sizeable proportion of targeted plasma membrane proteins do not accumulate internally, but are degraded in lysosomes, as indicated by rescue with the inhibitor leupeptin. The overlap that exists between US12 gene family targets with those of the KSHV K5 viral E3 ubiquitin ligase (MICA, MICB, B7 family, ALCAM, IFNGR1, PTPRM, EPHA2, CD99, MPZL1/2) could represent convergent evolution and/or targeting of similar cellular pathways (*Timms et al., 2013*).

The US12 family has a major impact on NK cell recognition with four US12 family members US12, US14, US18 and US20 consistently contributing toward significant levels of NK cell protection. Although the HCMV US21 deletion mutant also scored in functional assays, this could be due to enhanced expression of pUS20 associated with this construct. The NKG2DL MICB and ULBP2 are strongly upregulated during HCMV infection, but fail to reach the cell surface as they are retained in the ER by the NK evasion function gpUL16 (*Rolle et al., 2003*). Our proteomic data reveal that deletion of US12, US13 and US20 leads to increased cell surface and intracellular expression of MICB, ULBP2 and UL16 itself (*Figure 9*). These observations are consistent with US12 family members acting in concert to direct UL16, MICB and ULBP2 towards proteolytic degradation, with leupeptin treatment clearly able to rescue MICB, and this may contribute to the NK evasion properties of US12 (*Figure 1*). They also imply that some of the effects of the US12 family could be the result of co-operation with other HCMV genes. There is some precedent for this situation, although not previously on such a wide scale, as co-operation between US2 and UL141 was observed in the proteasomal degradation of some protein targets e.g. CD112 (*Hsu et al., 2015*). None of the US14 targets are recognized NK cell ligands, suggesting that this gene product utilizes a novel NK evasion mechanism.

B7-H6 is the major ligand identified for the natural cytotoxicity receptor (NCR) NKp30 and a tumor antigen (*Brandt et al., 2009*). Previously, B7-H6 expression was shown to be induced by inflammatory stimuli, for example pro-inflammatory cytokines and bacterial-derived toll-like receptor (TLR) ligands (*Matta et al., 2013*). We show here that B7-H6 expression is induced as a stress protein by HCMV infection (and potentially other virus infections) and that US18 and US20 act together to suppress cell surface expression of B7-H6 in the context of HCMV, thereby inhibiting NK cell activation. US18 and US20 were also able to regulate exogenously expressed B7-H6 when expressed individually. Therefore, they appear to target B7-H6 directly and not cellular pathways leading to the expression of B7-H6. The control of B7-H6 is likely to be of major significance as NKp30 is expressed on γδ T-cells (Vδ2⁻) that are induced during HCMV infection post-transplantation and correlate with control of disease (*Merville et al., 2000*; *Lafarge et al., 2001*; *Correia et al., 2011*).

The US12 family clearly impacts on a broad range of cellular functions including adhesion molecules and cytokine receptors. Many of these adhesion molecules play roles in co-stimulation or immune synapse formation of T-cells e.g. ALCAM, ICOSLG, CXADR (*Hassan et al., 2004*; *Wang et al., 2000*; *Witherden et al., 2010*). The US12 family targets pro-inflammatory mediators, such as multiple members of the tumour necrosis factor receptor (TNFR) superfamily (TNFRSF8, TNFRSF12A, NGFR, LTBR), gp130 (IL6ST), the IL-6 receptor signaling receptor, pannexin-1 (PANX1) and the TLR4 ligand high-mobility group protein B1 (HMGB1) (*Croft et al., 2013*; *Kanneganti et al., 2007*; *Park et al., 2004*). IL6ST signaling may also have antiviral effects in the context of HCMV (*Harwardt et al., 2016*), which was suggested by the finding that disruption of gp130 STAT3 binding resulted in IFN-like signaling through STAT1 (*Costa-Pereira et al., 2002*).

The closest cellular homologues to the US12 family are the TMBIM family with recognised functions in controlling apoptosis, ER stress, ROS production, actin production, glucose metabolism and protein trafficking (*Rojas-Rivera and Hetz, 2015*). The US12 family may have arisen through an 'accordion' gene expansion from a 'captured' ancestor TMBIM gene. Out of the entire US12 family, US21 displays the highest level of homology with TMBIM1/4, with the homologous region limited to the transmembrane domain and loops (unpublished observations, [*Lesniewski et al., 2006*; *Holzerlandt et al., 2002*]). The divergence of the US12 and TMBIM families is also reflected in their different membrane topologies, as TMBIM family have cytosolic N- and C-termini with six full transmembrane spanning regions, and US20 has a cytosolic N-terminus and lumenal C-terminus (*Carrara et al., 2012*; *Cavaletto et al., 2015*). The functional relevance of the US12 family/TMBIM homology remains unclear, although it may represent a 'functional scaffold' (*Lesniewski et al., 2006*), as a number of TMBIMs also target proteins for lysosomal degradation (*Lee et al., 2012*; *Yamaji et al., 2010*).

Our data indicate that the US12 family targets multiple host immune ligands, consistent with the family having arisen, been selected and diverged in function as a consequence of immune selection. US12 family genes differ from the majority of characterised immune evasion viral genes, as they act together and do not exhibit single gene effects. The targeting of multiple proteins by a HCMV immunevasin or co-operation between HCMV gene products is not unprecedented, but was not previously observed on this scale or with this diversity of target proteins (*Tomasec et al., 2005*; *Prod'homme et al., 2010*; *Smith et al., 2013*; *Hsu et al., 2015*). The majority of US12 family targets contain an immunoglobulin-like domain, including the MHC I-related proteins, MICA and B7-H6. Our data highlight the importance that HCMV gene families are likely to play in terms of HCMV persistence in vivo and identifies the US12 family as a critical region for regulation of the host immune response.

## Materials and methods

### Ethics statement
Healthy adult volunteers provided blood for this study following written informed consent (approved by the Cardiff University School of Medicine Ethics Committee Ref. no: 10/20) or buffy coats provided by the Welsh Blood Service, following informed consent.

### Cell lines
Human foreskin fibroblasts (HFFs), HFFs immortalized by human telomerase (HF-TERT), HF-TERTs transfected with the Coxsackie-adenovirus receptor (HF-CAR), were maintained at 37°C in 5% $CO_2$ in growth medium (Dulbecco's minimal essential medium (DMEM) supplemented with penicillin/streptomycin and 10% fetal calf serum (Invitrogen, Paisley, UK) (*McSharry et al., 2001*).

### Viruses
HCMV deletion mutants were generated by recombineering of the bacterial artificial chromosome (BAC) of HCMV strain Merlin (GenBank accession number GU179001.1), as described previously (*Stanton et al., 2010*). Strain Merlin contains the complete genetic complement of HCMV, and is frame shifted in two genes ($RL13^-$, $UL128^-$). Alterations to the BAC were monitored by local PCR and Sanger sequencing, and the entire genomes of the viruses were confirmed by Illumina sequencing as described previously (*Murrell et al., 2016*). Details of the viruses are provided in *Table 1* and a list of primers used in their construction and local sequencing of the BACs are provided in *Tables 2* and *3* respectively. Primers used in the construction of the HCMV △US18 and △US20 deletion mutants were detailed previously (*Fielding et al., 2014*).

The replication-deficient adenovirus vector (RAd-CTRL, pAL1253) and adenovirus vectors expressing US18 (RAd-US18) and US20 (RAd-US20) from the Merlin strain of HCMV have been described previously (*Fielding et al., 2014*). A B7-H6 (NCR3LG1) expressing adenovirus was generated by synthesising the B7-H6 CDS (corresponding to Accession number NM_001202439.2, bases 209–1573 retaining the stop codon) with 5' and 3' arms of homology to the AdZ BAC (corresponding to the For and Rev primers used in the AdZ recombineering protocol) (*Stanton et al., 2008*) and flanking EcoRI sites (gene synthesis from Genscript, Piscataway, NJ). The cassette containing the B7-H6 CDS and both arms of homology was released from the pUC57 vector by EcoRI digest, gel purified and inserted into pAL1141 by recombineering to produce pAL1593. BAC DNA of pAL1593 purified by maxiprep (Macherey-Nagel, Dueren, Germany) was transfected into 293 TREX to generate the adenovirus, as previously described (*Stanton et al., 2008*).

### HCMV infections
For proteomic analysis, 24 hr prior to each infection $1.5 \times 10^7$ HFFs were plated in a 150 $cm^2$ flask. Cells were sequentially infected at multiplicity of infection 10 with HCMV strain Merlin >90% of cells were routinely infected using this approach as assessed by MHC I down-regulation. Infections were staggered such that all flasks were harvested simultaneously. For NK cell degranulation, flow cytometry and western blot analyses, cells were seeded in growth medium at appropriate cell densities ($1 \times 10^6$ cells for a 25 $cm^2$ flask). The following day, the cells were infected with virus at the required multiplicity of infection in an appropriate volume of growth medium (2 ml for a 25 $cm^2$ flask) for 2 hr

**Table 2.** Primers used in the construction of the HCMV US12 deletion mutants library.

| Primer | Sequence (5' > 3') |
|---|---|
| US12 GalK For | GCGGGGGACAAAGGACAGTACGACACAGATTAGGTGATAGAAACGTTTTTTTCCTGTTGACAATTAATCATCGGCA |
| US12 GalK Rev | AAACTTGCCGGGTACCTGAAGCCCCGACGACTGTTCGTCGAGCACCCG TCTCAGCACTGTCCTGCTCCTT |
| US12 Delete | GCGGGGGACAAAGGACAGTACGACACAGATTAGGTGATAGAAACGTTTTTTTTGACGGTGCTCGACGAACAGTCGTCGGGGGCTTCAGGTACCCGGCAAGTTT |
| US13 GalK For | CTTCAGGTACCCGGCAAGTTTTATAGAGAAAGGGGACGATGGGTGGTGCCTGTTGACAATTAATCATCGGCA |
| US13 GalK Rev | GAAGACTCCACCGAGACGCTCACCCGTTCACCTGGGCGCATCACCCGC CTTCAGCACTGTCCTGCTCCTT |
| US13 Delete | CTTCAGGTACCCGGCAAGTTTTATAGAGAAAGGGGACGATGGGTGGTGAGGCGGGTGATGCGCCGAGTGAACAGGGTGAGCGTCTCGGGTGGAGTCTTC |
| US14 GalK For | GAGTGAACGGGTGAGCGTCTCGGTGGAGTCTTCTTATAAACCAGCGGG TCCCTGTTGACAATTAATCATCGGCA |
| US14 GalK Rev | CTGTAGCTTCGAGACCTTGCGGATACGCCGCCGGGCGTCCCG ACTCAGCACTGTCCTGCTCCTT |
| US14 Delete | GAGTGAACGGGTGAGCGTCTCGGTGGAGTCTTCTTATAAACCAGCGGGACCGGAGCGCAGCGCCGGGGCGTATCCGCAAGGTCTCGAAGCTACAG |
| US15 GalK For | CTCCATGTCGGGACCGCAGCGCCCGGCGTATCCGCAAGGTCTCGAAGCCTGTTGACAATTAATCATCGGCA |
| US15 GalK Rev | CGGAACTGGTTTTCGGGACCAGAGCAGCCGTTTCCAGAGAACGCAGCGCA CCTCAGCACTGTCCTGCTCCTT |
| US15 Delete | CTCCATGTCGGGACCGCAGCGCCCGGCGTATCCGCAAGGTCTCGGAAACGGCTGCGTTCTCTGGAAACGGCTGCTCTGTCCGAAAACCAGTTCCG |
| US16 GalK For | CGTTCTCTGGAAACGGCTGCTCTGTCCGAAAACCAGTTCCGAACGAAAATCCTGTTGACAATTAATCATCGGCA |
| US16 GalK Rev | CCCCACGGATCTCGCGTCTTAGACGCGGTCATATAGCCTCCGGCTG TCTCAGCACTGTCCTGCTCCTT |
| US16 Delete | CGTTCTCTGGAAACGGCTGCTCTGTCCGAAAACCAGTTCCGAACGAAAATGACAGCCGGAGGCTATATGACCGCGGTCTAAGACGCGGAGATCCGTGGGG |
| US17 GalK For | TTGGTGGAGACGGCCGGCGCGGGGGAAACGACGAGTTTTTCCGGACGCGGTATCAAAAGGCGCTATCAAAAGGCGCGGTGGCTATCAAAAGGCGCGGTATGAGAAACCGTTTATAGAGTGT |
| US17 GalK Rev | ACACTCTATAAACGGTTTCTACATACGCGCCTTTGATAGCCACCGCCG TCTCAGCACTGTCCTGCTCCTT |
| US17 Delete | TTGGTGGAGACGGCCGGCGCGGGGGAAACGACGAGTTTTTCCGGACGCGGTATCAAAAGGCGCGGTGGCTATCAAAAGGCGCGGTATGAGAAACCGTTTATAGAGTGT |
| US19 SacB For | CAGCACCCGGTTACCGCGGATTTGATTGACGTCACGAGTGTGGTCAAACCGTGGCGGCACCCGTATCCGACCCGTGTGACGGAAGATCACTTCG |
| US19 SacB Rev | GCTACGCCCTCTATGTCGAAAATGTGCTTTATTCATCGGCATGTACCATCTCTGGTTGTGGAGCCCATGACTGAGGTTCTTATGGCTCTTG |
| US19 Delete | ACGTCACGAGTGTGGTCAAACCGTGGCGGCACCCGTATCCGACCCGTGTGAAAACGGGCGCGGTTTTATAGGCATTAG |
| US21 SacB For | TGCGGCGCACCTACCCTTCTCTTATACACAAGCGAGCGAGTGGGGCACGGTGGTCACGCGCGGACACGTCGTGTGACGGAAGATCACTTCG |
| US21 SacB Rev | CAGCGCCCACACTGCTCAGACGCGGTCGCTGCGACGGTCGCCGCCCCAGTTCGTCTCCTAACTGAGGTTCTTATGGCTCTTG |
| US21 Delete | CAAGCGAGCGAGTGGGCACGGTGACGGTGACGTCACGCGCCGCGGACACGTCGAGGGCGGCAACGCCGCGGTTATCGCCGAGATTCGTCTAAATACACGAAGCG |
| US12-21 Delete | GCGGGGGACAAAGGACAGTACGACACAGATTAGGTGATAGAAACGTTTTTTTGGGCGCAACGCCGCGGTTATCGCCGAGATTCGTCTAAATACACGAAGCG |

**Table 3.** Primers used in the local sequencing of the HCMV US12 deletion mutant library

| Primer | Sequence (5' > 3') |
| --- | --- |
| US12 Seq For | CCCTGTCTAGACTCAAAAGCTG |
| US12 Seq Rev | ATCGTCCCCCTTTCTCTATA |
| US13 Seq For | GCCGAGTGGCTCGCC |
| US13 Seq Rev | CTGGGCACCTATCATCATTA |
| US14 Seq For | GGAGGGAAGCCCATTGC |
| US14 Seq Rev | TCATTACCTGTCTAGCCG |
| US15 Seq For | CGGACGCGGCTTCC |
| US15 Seq Rev | GTCGCTACAGCTCTTTATTA |
| US16 Seq For | GGGGCACGTAGATGACCG |
| US16 Seq Rev | CTCATTAGACAAACTCATCG |
| US17 Seq For | GTCTAAGACGCGAGATCCG |
| US17 Seq Rev | CCCAGTAGACAGACAGAACA |
| US19 Seq For | GGAGCGGCACGATGGTGACC |
| US19 Seq Rev | TCTGCCCACCTAACCAATGC |
| US21 Seq For | GCTGAAAGATGAAGATGGCG |
| US21 Seq Rev | ACCCGACCAGATGGGAGACG |

on a rocker at 37°C in 5% $CO_2$. The innoculum was then replaced with fresh growth medium (7 ml for a 25 cm$^2$ flask), and the cells were incubated for the required times. Fetal calf serum was omitted from the growth medium for mock and HCMV infections. For inhibitor studies, cells were treated 12 hr prior to harvesting with lysosomal inhibitors (leupeptin 200 µM, Merck Millipore, Watford, UK; Cat. no. 108975) in DMEM. For siRNA experiments, cells were seeded in a 25 cm$^2$ flask at $8 \times 10^5$ cells/flask 24 hr prior to transfection and then transfected in Optimem medium (Invitrogen) with 120 pmol B7-H6 (SI04761351, Hs_DKFZp686O24166_5, Qiagen, Manchester, UK ) or control siRNA (All-Star Negative Control siRNA, 1027281, Qiagen) using Lipofectamine RNAiMax (Invitrogen) for a further 24 hr before infection with HCMV (MOI 20) in serum free DMEM for a further 72 hr.

## Adenovirus infections

Adenovirus infections were carried out in HF-CAR (MOI 5). Cells were seeded in growth medium at appropriate cell densities ($1 \times 10^6$ cells for a 25 cm$^2$ flask). The following day, the cells were infected with virus at the required multiplicity of infection in an appropriate volume of growth medium (2 ml for a 25 cm$^2$ flask) for 2 hr on a rocker at 37°C in 5% $CO_2$. The inoculum was then replaced with fresh growth medium (7 ml for a 25 cm$^2$ flask), and the cells were incubated for the required times.

## Protein isolation, peptide labeling with tandem mass tags

Preparation of PM and WCL protein and peptide samples was performed as described previously (*Weekes et al., 2014*). For PM analysis, 100% of each tryptic peptide sample was labeled with TMT reagent, and 6 fractions generated from combined peptide samples by tip-based strong cation exchange. For WCL analysis, cells were lysed in 6 M Guanidine/50 mM HEPES pH8.5 then processed as described (*Weekes et al., 2014*). Proteins were digested with LysC then Trypsin. Peptides were labeled with TMT reagent, and 12 fractions generated by high pH reversed phase HPLC.

## Mass spectrometry and data analysis

Mass spectrometry and data analysis were performed as described previously (*Weekes et al., 2014*). Briefly, we performed mass spectrometry using an Orbitrap Fusion, and quantified TMT reporter ions from the MS3 scan (*McAlister et al., 2012*; *Ting et al., 2011*). Peptides were identified and quantified using a Sequest-based in-house software pipeline. A combined database was searched, consisting of: (a) human Uniprot, (b) Merlin strain HCMV Uniprot and (c) all additional

novel HCMV ORFs (*Stern-Ginossar et al., 2012*). Peptides spectral matches (PSM) were filtered to a 1% peptide false discovery rate (FDR) using linear discriminant analysis (*Huttlin et al., 2010*). The resulting dataset was further collapsed to a final protein-level FDR of 1%. Protein assembly was guided by principles of parsimony. Where all PSM from a given HCMV protein could be explained either by a canonical gene or novel ORF, the canonical gene was picked in preference. Proteins were quantified by summing TMT reporter ion counts across all matching PSM after filtering based on isolation specificity (*Pease et al., 2013*). Reverse and contaminant proteins were removed, and protein quantitation values were exported for normalization and further analysis in Excel. Where all PSMs from a given HCMV protein could be explained either by a canonical gene or novel ORF, the canonical gene was picked in preference. For five viral proteins that had related novel ORFs (N-terminal extensions of the viral protein), some peptides could either have originated either from the canonical protein or the novel ORF. In these cases, each of the novel ORFs were quantified based only on unique peptides that could only have originated from that ORF. Peptides that could either have originated from the canonical protein or the novel ORF were assigned to the canonical protein. We estimated p values for the ratios of each mutant compared to HCMV Merlin or leupeptin-treated cells infected with HCMV Merlin to HCMV Merlin-infected cells, using Benjamini-Hochberg corrected Significance B values (*Cox and Mann, 2008*). Hierarchical clustering was performed using centroid linkage with Pearson correlation. The Database for Annotation, Visualization and Integrated Discovery (DAVID, RRID:SCR_001881) was used to determine protein family enrichment amongst KEGG pathways (*Huang et al., 2009*). A given cluster was always searched against a background of all proteins quantified within the relevant experiment.

The mass spectrometry proteomics data have been deposited to the ProteomeXchange Consortium (*Vizcaíno et al., 2014*) via the PRIDE partner repository (*Vizcaíno et al., 2016*) with the dataset identifier PXD005883.

## Flow cytometric analysis of cell surface marker expression

Flow cytometry was performed as described previously (*Fielding et al., 2014*), except HF-TERTs were harvested using HyQtase (Thermo Fisher Scientific, Paisley, UK) for 3 min at 37°C, instead of Trypsin/EDTA. The following phycoerytherin (PE)-conjugated antibodies were used at the indicated dilutions (200 μl per stain): anti-CD166/ALCAM (Biolegend, London, UK, Clone 3A6, Cat. no. 343904, 1:80, RRID:AB_2289302), anti-EPHA2 (Biolegend, Clone SHM16, Cat. no. 356804, 1:80, RRID:AB_2561807), anti-CD323/JAM3 (Biolegend, Clone SHM33, Cat. no. 356704, 1:80, RRID:AB_2561802), anti-CD130/IL6ST (Biolegend, Clone 2E1B02, Cat. no. 362003, 1:80, RRID:AB_2563401), IgG1 isotype (Biolegend, Clone MOPC-21, Cat. no. 400112, 1:80, RRID:AB_326434), mouse IgG2a isotype (Biolegend, Clone MOPC-173, Cat. no. 400212, 1:160, RRID:AB_326460) and mouse IgG2b isotype (Biolegend, Clone MPC-11, Cat. no. 400314, 1:80, RRID:AB_326492). The following unconjugated antibodies were used at the indicated dilutions (200 μl per stain): anti-CXADR/CAR (Merck Millipore, Cat no. 05–644, Clone RmcB, 1:500, RRID:AB_309871), anti-CD266/TWEAK R/TNFRSF12A (Biolegend, Clone ITEM-4, Cat. no. 314102, 1:200, RRID:AB_2240752), anti-MICA (BAMOMAB, Graefelfing, Germany, Cat. no. AM01, 1:400, RRID:AB_2636811), anti-MICB (BAMOMAB, Cat. no. BM02, 1:400, RRID:AB_2636812), anti-B7-H6 (Biotechne R and D Systems, Abingdon, UK, Clone 875001, Cat. no. MAB7144, 500 μg/ml, 1:50, RRID:AB_2636810), anti-MHC I (BioRAd/AbD Serotec, Kidlington, UK, Clone W6/32, MCA81EL, 1:1000, RRID:AB_324063) and mouse IgG (Santa Cruz Biotechnology, Heidelberg, Germany, Cat. no. Sc-2025, 400 μg/ml, 1:40, RRID:AB_737182 or Sigma Aldritch, Gillingham, UK, Cat. no. I-5381 1 mg/ml, 1:100, RRID:AB_1163670), followed by an Alexa Fluor 647 goat anti-mouse IgG secondary antibody (Thermo Fisher Scientific, Cat. no. A21237, 1:500, RRID:AB_2535806).

## Immunoblotting

Whole cell lysates were prepared in 1x Nupage gel sample buffer (Thermo Fisher Scientific) plus 10 mM DTT and samples were denatured at 95°C or 50°C (US18 and US20 adenovirus experiment) for 10 mins. SDS-PAGE and subsequent immunoblotting was carried out as previously described either using X-ray film or a G:Box Chemix-xx6 GeneSys system (Syngene, Cambridge, UK) to visualise the blots (*Fielding et al., 2014*). Membranes were probed with antibodies directed against B7-H6 (non-commercial CH31 monoclonal, purified at 2 μg/ml final or hybridoma supernatant 1:5), anti-V5

antibody (BioRad/AbD Serotec, Cat. no. MCA1360, 1:2000, RRID:AB_322378) and actin (Sigma Aldritch, Cat. no A2066, 1:5000, RRID:AB_476693), followed by HRP-conjugated goat anti-mouse or anti-rabbit antibodies (BioRad/Ab Serotec, Cat. no. 170–6516, RRID:AB_11125547 and 170–6515, RRID:AB_11125142 respectively, both 1:5000).

## NK degranulation assays

NK degranulation assays were performed as described previously using anti-CD107a-FITC (Cat. no. 555800, BD Biosciences, Oxford, UK, RRID:AB_396134) or isotype control-FITC (Cat no. 555748, BD Biosciences, RRID:AB_396090) and anti-CD3-PE-Cy7 (Cat. no. 737657, Beckman Coulter, High Wycombe, UK RRID:AB_2636813) and anti-CD56-PE (Cat. no. A07788, Beckman Coulter, RRID:AB_2636814) antibodies and PBMC derived from buffy coats or donor blood, except that infected HF-TERTs were harvested using HyQtase for 3 min at 37°C, instead of Trypsin/EDTA (*Prod'homme et al., 2007*, *2010*; *Fielding et al., 2014*). Blocking experiments were performed as previously described except using B7-H6 blocking antibody (CH31) or isotype IgG1 control antibody at 10 μg/ml (Biolegend, Cat. no. 401404, Clone MG1-45, RRID:AB_345426 or Biolegend, Cat no. 401402 Clone MG1-45, RRID:AB_345424) (*Fielding et al., 2014*).

## NKp30 reporter assays

HF-TERTs were seeded into 96-well plates and either mock-infected or infected with HCMV or US12 family deletion mutants (MOI 10) for 72 hr. The CT299 reporter line (2B4 cells stably transfected with an NFAT-GFP reporter and NKp30) was washed in complete RPMI and 50,000 reporter cells added per well (5:1 ratio of reporter to target cells). In some experiments, the B7-H6 blocking antibody (CH31) or isotype IgG1 control antibody (Biolegend, Cat. no. 401404, Clone MG1-45, RRID:AB_345426 or Biolegend Cat no. 401402 Clone MG1-45, RRID:AB_345424) at 10 μg/ml were also added to the wells. After 24 hr incubation, reporters were harvested, washed in FACS buffer and fixed with 2% paraformaldehyde before analysis by flow cytometry.

## Acknowledgements

This work was supported by funds from the Wellcome Trust WT090323MA and MRC G1000236. MR/L018373/1. This work was also supported by a Wellcome Trust Principal Research Fellowship (WT101835) to PJL and a Wellcome Trust Senior Fellowship (108070/Z/15/Z) to MPW and a strategic award to Cambridge Institute for Medical Research from the Wellcome Trust (100140). JAP was supported by NIH/NIDDK grant K01 DK098285. BV was supported by projects MEYS – NPS I – LO1413 and Czech Science Foundation P206/12/G151. ER was supported by MH CZ-DRO (MMCI, 00209805). CC and JT were supported by grant G0901682 from the MRC and funding from the European Research Council (ERC) under the European Union's Horizon 2020 research and innovation programme (grant agreement No 695551), with partial funding from the National Institute of Health Cambridge Biomedical Research Centre. The authors are grateful to Dawn Roberts for technical support and to blood donors who contributed to the study.

## Additional information

### Funding

| Funder | Grant reference number | Author |
|---|---|---|
| Wellcome | 108070/Z/15/Z | Michael P Weekes |
| National Institute of Diabetes and Digestive and Kidney Diseases | K01 DK098285 | Joao A Paulo |
| Czech Science Foundation | P206/12/G151 | Chiwen Chang |
| European Research Council | 695551 | Borek Vojtesek |
| Medical Research Council | G0901682 | John Trowsdale |
| Medical Research Council | MRC G1000236 | Peter Tomasec Gavin W G Wilkinson |

| Wellcome | WT090323MA | Peter Tomasec Gavin W G Wilkinson |
|---|---|---|
| Medical Research Council | MR/L018373/1 | Peter Tomasec Gavin W G Wilkinson |
| Wellcome | WT101835 | Paul J Lehner |
| Wellcome | 100140 | Paul J Lehner |

The funders had no role in study design, data collection and interpretation, or the decision to submit the work for publication.

## Author contributions

CAF, Conceptualization, Resources, Data curation, Software, Formal analysis, Supervision, Validation, Investigation, Methodology, Writing—original draft, Project administration, Writing—review and editing; MPW, Conceptualization, Data curation, Software, Formal analysis, Supervision, Validation, Investigation, Visualization, Methodology, Writing—original draft, Project administration, Writing—review and editing; LVN, Resources, Data curation, Software, Formal analysis, Visualization, Writing—original draft; ER, CC, RJS, RJA, BV, JT, Resources, Methodology; GSW, Data curation, Formal analysis, Validation, Writing—original draft; JAP, Resources, Data curation, Software, Formal analysis; NMS, Data curation, Software, Formal analysis, Validation, Writing—original draft; JAD, HN, Resources, Investigation; RA, Data curation, Investigation; AJD, Resources, Data curation, Software, Formal analysis, Methodology, Writing—original draft; SPG, Resources, Data curation, Software, Formal analysis, Methodology; PT, GWGW, Conceptualization, Supervision, Investigation, Methodology, Writing—original draft, Project administration, Writing—review and editing; PJL, Conceptualization, Resources, Data curation, Software, Supervision, Writing—original draft, Project administration, Writing—review and editing

## Author ORCIDs

Ceri A Fielding, http://orcid.org/0000-0002-5817-3153
Michael P Weekes, http://orcid.org/0000-0003-3196-5545
Luis V Nobre, http://orcid.org/0000-0003-0467-8989
Joao A Paulo, http://orcid.org/0000-0002-4291-413X
Nicolás M Suárez, http://orcid.org/0000-0001-8429-8374
Robin Antrobus, http://orcid.org/0000-0001-8608-4011
Richard J Stanton, http://orcid.org/0000-0002-6799-1182
Hester Nichols, http://orcid.org/0000-0002-6814-4364
John Trowsdale, http://orcid.org/0000-0002-0150-5698
Andrew J Davison, http://orcid.org/0000-0002-4991-9128
Peter Tomasec, http://orcid.org/0000-0002-6745-6198
Paul J Lehner, http://orcid.org/0000-0001-9383-1054
Gavin W G Wilkinson, http://orcid.org/0000-0002-5623-0126

## Ethics

Human subjects: Healthy adult volunteers provided blood for this study following written informed consent (approved by the Cardiff University School of Medicine Ethics Committee Ref. no: 10/20) or buffy coats provided by the Welsh Blood Service, following informed consent.

## Additional files

### Supplementary files

• Supplementary file 1. Interactive spreadsheet of all US12 family proteomic data (separate Excel Spreadsheet). This spreadsheet enables the generation of graphs showing the relative expression of any of the human and HCMV proteins quantified across Proteomic series 1–3 in PM and WCL fractions by typing the gene name into the indicated box. The datasets used to generate the graph and protein aliases are present in the other tabs.

## Major datasets

The following dataset was generated:

| Author(s) | Year | Dataset title | Dataset URL | Database, license, and accessibility information |
|---|---|---|---|---|
| Michael P Weekes | 2017 | Control of immune ligands by members of a cytomegalovirus gene expansion suppresses natural killer cell activation | http://www.ebi.ac.uk/pride/archive/projects/PXD005883 | Publicly available at the PRIDE archive (accession no. PXD005883) |

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
