## [Decision Letter]

Thank you for submitting your article "Control of immune ligands by members of a cytomegalovirus gene expansion suppresses natural killer cell activation" for consideration by *eLife*. Your article has been favorably evaluated by Wenhui Li (Senior Editor) and three reviewers, one of whom is a member of our Board of Reviewing Editors. The following individual involved in review of your submission has agreed to reveal his identity: Michael G Brown (Reviewer #2).

The reviewers have discussed the reviews with one another and the Reviewing Editor has drafted this decision to help you prepare a revised submission.

Summary:

Human cytomegalovirus (HCMV) devotes a significant portion of its dsDNA genome to produce molecules that improve "viral fitness," most often resulting in immune evasion. Here Fielding et al. focus on a region of the HCMV genome, termed US21 family (US12-US21). They took a comprehensive mass spectrometry approach to determine which molecules are affected by deleting these viral genes, using single and complete US21 family knockout viruses. The data strongly support the overall conclusion that these molecules affect NK cell activation against virus-infected cells. That the US21 family affects NKG2D ligands is not surprising but they also demonstrate nicely that US18 and US20 affect expression of B7-H6, that is induced upon viral infection with mutants lacking either one of these ORFs. Overall, the manuscript was favorably viewed but the reviewers felt that a number of issues need to be addressed.

Essential revisions:

While the comments of the reviewers are attached and should be addressed, the reviewers request that the authors specifically address the following major points in a revision.

1) The regulation of B7-H6 by ectopic expression of US18 and/or US20. See reviewers #1 and #3.

2) Explicit demonstration that US20 effect on MIC-A is impacted by US21. This is mentioned as data not shown but the reviewers would like the authors to show the data. See reviewer #3.

3) The% infected cells should be included in each of the figure legends. See reviewer #3.

4) The statistical analysis of the experiments. See reviewer #2.

*Reviewer #1:*

It would be informative to express US18 and US20 individually or together by transfection and determine their effects on B7-H6 expression. This will help to confirm that the data with the individual ko viruses are not due to some other effect on the viral genome, such as an inadvertent deletion of an unrecognized ORF in the ko viruses.

Figure 11 shows upregulation of B7-H6 expression by control adenovirus infection, helping to make the case that viral infection induces B7-H6 expression. I don't believe this has been shown before, though there are some data indicating that B7-H6 is inducible (Blood 2013, 122:394). However, neither of these points was mentioned in the text which instead focused on the markedly higher levels of B7-H6 expression when the adenoviral vector contained B7-H6. It would be informative if the adenoviral vector contained US18, US20, or both to determine if B7-H6 expression is down-regulated only when both ORFs are ectopically expressed.

Finally, it would be of interest to determine if cells constitutively expressing high levels of B7-H6 are affected by US18/US20. If not, the ORFs may be affecting the mechanisms leading to viral induction of B7-H6, rather than B7-H6 itself.

*Reviewer #2:*

Strengths of the study include its innovative approach, in depth analysis of US12 related genes and their broad effect on host immune responses/proteomes, and follow up analyses of specific protein effects (e.g. B7H6). The work is technically excellent, with results effectively presented. The color scheme used to present data for comparison of various HCMV strains, and their effect on many different host molecules has transformed what would appear to be very complex data sets, into ones that are both manageable and elegant. Having implicated specific US12-targeted effects on host proteins, and that these molecules have evolved tag-team strategies to thwart NK cells, is highly interesting. Overall, the findings and conclusions are openly presented, and very compelling. They should generate considerable interest. Although effective strategies and cutoffs were implemented to focus on findings of high significance, there is some question and lingering concern about the use of statistical tests applied, or not, in some experiments.

Main concern:

Multiple comparisons performed in Figure 1 using a student's t test may be inappropriate. Also, it appears that statistical analyses were not performed in Figure 4, Figure 5, Figure 7 and Figure 8. Why?

*Reviewer #3:*

Interestingly, and somewhat of concern, deletion of US21 impacted the expression of many other viral genes, including US20 whereas this was not observed when the entire region US12-21 was deleted. The authors argue that this is likely due to the fact that the impact on US20 by US21 no longer occurs when both are deleted, but this is not explicitly shown. I suggest these data are included upon revision.

Interestingly, they find that deletion of US12, US13 and US20 upregulates UL16 as well deletion of US12-21. Since UL16 prevents the intracellular transport of multiple NKG2D-ligands this result could imply that some of the observed changes in NK cell molecules could be an indirect consequence of US12 family members promoting the expression or stabilization of UL16 which in turn could decrease the lysosomal turnover of target proteins by retaining them in the ER. The possibility that the effects seen are indirect consequences of US12 proteins regulating other viral proteins should be acknowledged more explicitly in the text and it should be clearly stated that this is one of the caveats of the current study.

The authors next focus on one protein in particular, B7-H6, a ligand for the NK cell activating receptor NKp30 that is also found on γδT cells. Using adenoviral expression they demonstrate that B7-H6 activates NK cells. Furthermore, a B7-H6 dependent reporter cell line is activated by US18 and US20, as well as US12-21 deletions and this activation can be blocked by B7-H6 antibodies or siRNA. However, given the concerns above, it is quite possible that neither US18 or US20 directly impact B7-H6. This would be easy to test using the rAd-US18 and rAd-US20 previously described by the same group (Fielding 2014). Therefore, I suggest that an experiment is included that examines whether US18 and US20 affect B7-H6 expression directly upon ectopic expression.

For each experiment, the MHC-I downregulation results should be provided as supplemental data to demonstrate that a similar percentage of cells are infected when different mutants are compared.

---

## [Author Response]

*Essential revisions:*

*While the comments of the reviewers are attached and should be addressed, the reviewers request that the authors specifically address the following major points in a revision.*

*1) The regulation of B7-H6 by ectopic expression of US18 and/or US20. See reviewers #1 and #3.*

We have now included data showing targeting of exogenously expressed B7-H6 by US18 and US20, ectopically expressed via adenovirus vectors (panel D in Figure 11) – see response to reviewer #1, point 3 and reviewer #3, point 3).

*2) Explicit demonstration that US20 effect on MIC-A is impacted by US21. This is mentioned as data not shown but the reviewers would like the authors to show the data. See reviewer #3.*

In cells infected with the US21 deletion mutant, we observed increased expression of US20 by proteomics. We have included an additional figure (newly added Figure 7) showing these results (see detailed response to reviewer #3). Reviewer #3 did not directly mention subsequent effects of US20 upregulation on MICA, but any effect cannot be directly analysed in HCMV-infected cells, as infection with the HCMV control virus leads to complete down- regulation of MICA (The MICA signal is effectively zero in HCMV-infected cells).

*3) The% infected cells should be included in each of the figure legends. See reviewer #3.*

We have inserted% infected cells into the figure legends for Figure 1 (previous Figure 9) and 12F (see detailed response to reviewer #3).

*4) The statistical analysis of the experiments. See reviewer #2.*

An unpaired one-way ANOVA analysis of the data shown in Figure 1 has been performed and incorporated into the manuscript (see detailed response to reviewer #2). We have added p values into current Figure 4, Figure 5, Figure 7 (new figure), 8 (previous Figure 7) and 9 (previous Figure 8), estimated using Benjamini-Hochberg corrected Significance B values (Cox & Mann, 2008).

*Reviewer #1:*

*It would be informative to express US18 and US20 individually or together by transfection and determine their effects on B7-H6 expression. This will help to confirm that the data with the individual ko viruses are not due to some other effect on the viral genome, such as an inadvertent deletion of an unrecognized ORF in the ko viruses.*

We carried out whole genome sequencing of the virus genomes from all the US12 deletion mutants. The sequencing data has been submitted to Genbank and Accession numbers are shown in Table 1. This was stated in the text as follows: ‘Viruses were generated by DNA transfection and the complete genomic sequence of the virus stocks was validated by deep sequencing (Table 1).’ There were no changes found in the genome of the US18 and US20 deletion mutants, except for the targeted gene deletions.

The viral transcriptome of the HCMV Merlin strain has been analyzed by a number of groups (Gatherer et al., PNAS 108:19755 2011; Stern-Ginossar et al., Science338:1088 2013).

Ribosomal profiling identified a number of novel ORFs (Stern-Ginossar et al., 2013). Deletion of the US20 ORF is not predicted to affect any other identified ORFs and deletion of the US18 ORF would be predicted to delete only one single novel ORF (ORFS346CiORF1). This utilises an alternative ATG in the same reading frame as US18 encoding a protein corresponding to the C-terminal 37 amino acids of US18. We know of no evidence for other novel ORFs encoding proteins that might be affected by deletion of US20 or US18.

We have also performed experiments expressing US18 and US20 individually and together using adenovirus expression vectors, which indicate these proteins are capable of down- regulating B7-H6 levels when expressed on their own outside of the context of HCMV (see response to reviewer #1 point 3 and reviewer #3 point 3).

*Figure 11 shows upregulation of B7-H6 expression by control adenovirus infection, helping to make the case that viral infection induces B7-H6 expression. I don't believe this has been shown before, though there are some data indicating that B7-H6 is inducible (Blood 2013, 122:394). However, neither of these points was mentioned in the text which instead focused on the markedly higher levels of B7-H6 expression when the adenoviral vector contained B7-H6.*

The histogram shown in Figure 11 does not show mock-infected cells, but compares staining levels of the control adenovirus vector with the B7-H6-expressing adenovirus vector using a control murine IgG and B7-H6 specific antibody. We have not investigated the specific regulation of B7-H6 by control adenovirus vector infection. We have altered a line in the Results section to ‘B7-H6 is therefore induced as a ‘stress ligand’ (subsection “B7-H6 is an NK cell activating ligand targeted by US18 and US20”, first paragraph) and added a comment regarding the induction of B7-H6 to the Discussion, ‘…Previously, its expression was shown to be induced by inflammatory stimuli, for example pro-inflammatory cytokines and bacterial-derived toll-like receptor (TLR) ligands (Matta et al., 2013). We show here that B7-H6 expression is induced as a stress protein by HCMV infection (and potentially other virus infections)…’.

*It would be informative if the adenoviral vector contained US18, US20, or both to determine if B7-H6 expression is down-regulated only when both ORFs are ectopically expressed.*

*Finally, it would be of interest to determine if cells constitutively expressing high levels of B7-H6 are affected by US18/US20. If not, the ORFs may be affecting the mechanisms leading to viral induction of B7-H6, rather than B7-H6 itself.*

We have now provided data showing a reduction in ectopically expressed B7-H6 levels by the US18 or US20 adenovirus vectors individually. The following sentence has been added to the Results section, ‘Ectopic expression of US20, and to a lesser extent US18, from adenovirus vectors reduced levels of exogenously expressed B7-H6 detected by immunoblotting (Figure 11).’

We have also modified a section in the Discussion that now reads ‘…US18 and US20 act together to suppress cell surface expression of B7-H6 in the context of HCMV, thereby inhibiting NK cell activation. US18 and US20 were also able to regulate exogenously expressed B7-H6 when expressed individually. Therefore, they appear to target B7-H6 directly and not cellular pathways leading to the expression of B7-H6.’

In addition, the US18 and US20 adenoviruses were referenced in the Viruses heading of the Materials and methods section.

*Reviewer #2:*

*Strengths of the study include its innovative approach, in depth analysis of US12 related genes and their broad effect on host immune responses/proteomes, and follow up analyses of specific protein effects (e.g. B7H6). The work is technically excellent, with results effectively presented. The color scheme used to present data for comparison of various HCMV strains, and their effect on many different host molecules has transformed what would appear to be very complex data sets, into ones that are both manageable and elegant. Having implicated specific US12-targeted effects on host proteins, and that these molecules have evolved tag-team strategies to thwart NK cells, is highly interesting. Overall, the findings and conclusions are openly presented, and very compelling. They should generate considerable interest. Although effective strategies and cutoffs were implemented to focus on findings of high significance, there is some question and lingering concern about the use of statistical tests applied, or not, in some experiments.*

Main concern:

*Multiple comparisons performed in Figure 1 using a student's t test may be inappropriate. Also, it appears that statistical analyses were not performed in Figure 4, Figure 5, Figure 7 and 8. Why?*

Figure 1: We have now performed unpaired one-way ANOVA analysis with Dunnett’s test for multiple comparisons against the HCMV control. The new analysis did alter the statistical significance of some of the changes in NK degranulation observed. We have incorporated results of the statistical analysis into the figure legend of Figure 1 and the text as follows. ‘Significantly increased levels of NK activation were detected in assays using deletion mutants of 5 different US12 family members: US12 (3 of 4 donors), US14 (1 of 4 donors, with a trend towards increased NK activation in the other 3 donors), US18 (3 of 4 donors), US20 (4 of 4 donors), US21 (4 of 4 donors) (Figure 1), while three US12 family deletion mutants (US15, US16, US19) reduced the level of NK cell activation in some donors (Figure 1).’

Estimated p values based on Benjamini-Hochberg corrected Significance B values (Cox & Mann, 2008) have been added to current Figure 4, current Figure 5, newly added Figure 7, current Figure 8 (previous Figure 7) and current Figure 9 (previous Figure 8). Figure legends have updated accordingly.

*Reviewer #3:*

*Interestingly, and somewhat of concern, deletion of US21 impacted the expression of many other viral genes, including US20 whereas this was not observed when the entire region US12-21 was deleted. The authors argue that this is likely due to the fact that the impact on US20 by US21 no longer occurs when both are deleted, but this is not explicitly shown. I suggest these data are included upon revision.*

We have inserted the data showing up-regulation of US20 and down-regulation of representative late proteins (UL32 and UL99) in the delta US21 mutant infected cells into a new figure, current Figure 7. The text present in the manuscript, which previously referred to these data as ‘data not shown’ now references Figure 7 (subsection “Functional independence and co-operation exerted by family members”, second paragraphs).

*Interestingly, they find that deletion of US12, US13 and US20 upregulates UL16 as well deletion of US12-21. Since UL16 prevents the intracellular transport of multiple NKG2D-ligands this result could imply that some of the observed changes in NK cell molecules could be an indirect consequence of US12 family members promoting the expression or stabilization of UL16 which in turn could decrease the lysosomal turnover of target proteins by retaining them in the ER. The possibility that the effects seen are indirect consequences of US12 proteins regulating other viral proteins should be acknowledged more explicitly in the text and it should be clearly stated that this is one of the caveats of the current study.*

We agree with the reviewer that the effects of US12, US13 and US20 on UL16 could imply the effect of the US12 family on NK ligands may be partly indirect, and that the US12 family could be working in co-operation with other HCMV-encoded immune evasins such as UL16. We do not agree that the US12 family members promote expression/stabilize UL16, since the levels of UL16 actually increase (rather than decrease) during infection with the specific US12 deletion mutants (current Figure 9). These data support a role for the US12 family in promoting lysosomal degradation of UL16. We recently reported a similar co-operation between US2 and UL141 (Hsu et al., 2015 PLoS Pathogens11(4):e1004811) where UL141 retains CD155 in the ER and selectively feeds CD112 to US2 for proteasomal degradation.

The alterations in levels of UL16 imply that several members of the US12 family are targeting UL16 for lysosomal degradation, together with any UL16-bound ligands. These data also support the idea that the US12 family works through co-operation not only with each other, but potentially with other HCMV proteins. We have inserted the following text into the Discussion, ‘They also imply that some of the effects of the US12 family could be the result of co-operation with other HCMV genes. There is some precedent for this situation as co-operation between US2 and UL141 was observed in the proteasomal degradation of some protein targets e.g. CD112 (Hsu et al., 2015).’

*The authors next focus on one protein in particular, B7-H6, a ligand for the NK cell activating receptor NKp30 that is also found on γδT cells. Using adenoviral expression they demonstrate that B7-H6 activates NK cells. Furthermore, a B7-H6 dependent reporter cell line is activated by US18 and US20, as well as US12-21 deletions and this activation can be blocked by B7-H6 antibodies or siRNA. However, given the concerns above, it is quite possible that neither US18 or US20 directly impact B7-H6. This would be easy to test using the rAd-US18 and rAd-US20 previously described by the same group (Fielding 2014). Therefore, I suggest that an experiment is included that examines whether US18 and US20 affect B7-H6 expression directly upon ectopic expression.*

As discussed above in the response to reviewer #1 point 3, we now provide data showing a reduction in ectopically expressed B7-H6 levels by the US18 or US20 adenovirus vectors individually. The following sentence has been added to the Results section, ‘Ectopic expression of US20, and to a lesser extent US18, from adenovirus vectors reduced levels of exogenously expressed B7-H6 detected by immunoblotting (Figure 11).’

We have also modified a section in the Discussion that now reads ‘…US18 and US20 act together to suppress cell surface expression of B7-H6 in the context of HCMV, thereby inhibiting NK cell activation. US18 and US20 were also able to regulate exogenously expressed B7-H6 when expressed individually. Therefore, they appear to target B7-H6 directly and not cellular pathways leading to the expression of B7-H6.’ In addition, the US18 and US20 adenoviruses were referenced in the Viruses heading of the Materials and methods section.

*For each experiment, the MHC-I downregulation results should be provided as supplemental data to demonstrate that a similar percentage of cells are infected when different mutants are compared.*

We have inserted information regarding MHC-I downregulation from experiments comparing different mutants into the figure legends for Figure 1 (previous Figure 9) and 12F. For reporter assay experiments, cells were infected in the 96-well plates used for the assay and infections were monitored by light microscopy for HCMV cytopathic effect prior to the assay.